# ENHANCING LLM FAITHFULNESS IN RATIONALE GENERATION VIA DUAL-REWARD PROBABILISTIC INFERENCE

## ABSTRACT

As large language models (LLMs) are increasingly applied to complex reasoning tasks, achieving both accurate task performance and faithful explanations becomes crucial. However, LLMs often generate unfaithful explanations, partly because they do not consistently adhere closely to the provided context. Existing approaches address this problem either rely on superficial calibration, such as decomposed Chain-of-Thought prompting, or require costly retraining to improve model faithfulness. In this work, we propose a probabilistic inference paradigm that provides fine-grained and lookahead rewards to ensure that LLM-generated rationales are logically coherent and comprehensive. These rewards are derived from a domain-specific proposal distribution, allowing for optimised sequential Monte Carlo approximations. Our evaluations across three different reasoning tasks show that this method, which allows for controllable generation during inference, improves both accuracy and faithfulness of LLMs while keeping computational costs similar to those of existing decoding techniques. This method offers a promising path towards making LLMs more reliable for reasoning tasks without sacrificing performance or efficiency.

## 1 INTRODUCTION

Large language models (LLMs) have achieved remarkable success across a wide range of challenging tasks, including Question Answering (QA) (Li et al., 2024b), reasoning (Yao et al., 2023; Yan et al., 2024) and providing feedback on essays or reviews (Liang et al., 2024; Li et al., 2023). However, the opaque nature of these models makes it difficult to generate faithful explanations for their decision-making processes. While LLMs can be prompted to generate self-explanations when making predictions (Kim et al., 2024; Madsen et al., 2024; Atanasova et al., 2023), ensuring the fidelity of these rationale remains challenging: both for improving interpretability and for enhancing reliability in safety-critical fields (Lyu et al., 2024).

Enhancing the faithfulness of LLM-generated rationales is a multifaceted challenge. To date, there is no universally accepted or formal definition of faithfulness (Lyu et al., 2024). In this paper, we focus on a specific category of unfaithfulness, where models fail to incorporate key contextual information into their generated rationales. It is motivated by faithfulness evaluation, where unfaithful models do not respond adequately to alterations in input (Lanham et al., 2023; Radhakrishnan et al., 2023). This example in Figure 1 highlights that the Llama3 tends to assign high probabilities to generic words, resulting in rationale that is less coherent with the given context. This limitation may contribute to potential unfaithfulness in the model's outputs. In contrast, an expert model (logits in red), specifically trained on a scientific corpus, significantly increases the likelihood of domain-specific words that align more closely with the context. Moreover, the results presented in Table 1 show that the Llama3 exhibits a weak correlation between its generated assessments and the content of the provided in student's reports, while the expert model show significant improvements.

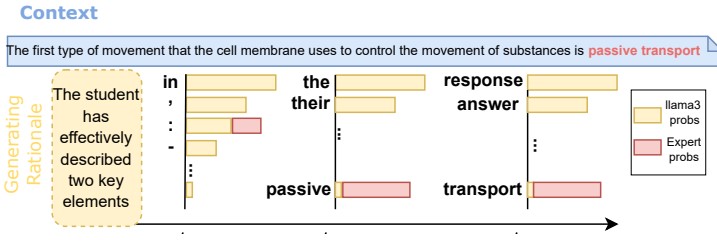

Figure 1: The decoding phrase in the untuned Llama3-8B-Instruct for assessing a student's answer to a biology question. **Llama3 tends to generate overly generic words**, such as '*the*' and '*response*', while **ignoring domain-specific words in the context**. The expert model is more sensitive to and utilizes such domain-specific words in context, such as '*passive*', '*transport*'.

| BLEU | Llama3 | Expert |
|---|---|---|
| 2-gram | 0.085 | 0.683 |
| 3-gram | 0.081 | 0.544 |
| 4-gram | 0.076 | 0.436 |

Table 1: The **semantics overlap** between the student reports in the *biology* subject and the LLM-generated assessment rationales. Llama3-8B exhibits an overall lower overlap with the given context compared to a domain-specific expert model, implying a tendency towards unfaithfulness.

The model's tendency to rely heavily on its pretrained distributions can be attributed to its ignorance of context, often causing it to overlook subtle differences across various domains and contexts (Hu et al., 2023). Evidence from multiple studies suggests that LLMs often generate inaccurate labels when applied to out-of-distribution scenarios (Yuan et al., 2023). Lin et al. (2024) showed that instruction-tuned LLMs perform almost identically with base model in decoding across most token positions, suggesting that even advanced LLMs struggle with adapting to new domains and generating contextually sensitive responses.

To improve the model's faithfulness by enhancing its sensitivity to context, we propose **a probabilistic inference paradigm** for generating faithful explanations. This approach incorporates *both local and anticipatory context coherence rewards* into a sequential Monte Carlo search. These rewards are inspired by the observation that LLMs struggle to generate domain-specific distributions due to their reliance on pre-trained data. Specifically, we introduce a step-wise filtering proposal distribution that adjusts the generation distribution of original LLMs. This optimised distribution stands out by (i) being more sensitive to domain-specific nuances; and (ii) accounting for future rewards during the generation process. Our contribution can be summarized as three-fold[1]:

- We investigate the challenge of faithful rationale generation by highlighting the limitations of general LLMs in producing domain-specific responses. To the best of our knowledge, this is the first study to enhance faithfulness by explicitly encouraging the generation of domain-specific tokens.
- We propose two novel reward mechanisms, namely local and global rewards, tailored for the faithfulness problem. These are integrated into a probabilistic inference framework to achieve a trade-off between task accuracy and rationale faithfulness.
- Empirical results with the Llama 3 backbone model show an absolute accuracy improvement of 33% over the seven datasets, along with a 10% improvements in faithfulness evaluation, while maintaining a computational cost similar to beam search (1.3×).

## 2    RELATED WORK

**Constrained decoding.** Constrained generation can be accomplished by fine-tuning, such as RLHF (Ouyang et al., 2022) and DPO (Rafailov et al., 2023). Further inspired by the observations that alignment can be achieved by searching and planning in the large decoding space without costly training, many recent papers have proposed probabilistic inference. Many are focused on optimization, such as beam search (Meister et al., 2020), others focus on sampling from a constrained or modified distribution, including naive rejection sampling (Poesia et al., 2022), nucleus sampling (Holtzman et al., 2020) and also GFlowNet targeting at more diverse distribution (Bengio et al., 2021). Our solution is to adopt a domain-specific proposal distribution to adjust the original posterior.

---

[1]We will open-source our code on GitHub upon acceptance.

**Faithful rationale.** Although LLMs can provide plausibly sounding explanations for their answers, recent work argues that model generated natural language explanations are often unfaithful (Lanham et al., 2023; Atanasova et al., 2023). Faithfulness evaluation for rationale is to apply important perturbation to the original rationale and check the changes in the new output. Such perturbation includes counterfactual edit (Atanasova et al., 2023), biased feature (Turpin et al., 2023) and corrupted Chain-of-Thought (Lanham et al., 2023). To increase the faithfulness of LLM-generated response, many existing methods focus on the Chain-of-Thought and decompose the reasoning process into multiple sub-sentences (Radhakrishnan et al., 2023), then verify them using external tool, e.g., python interpreter (Lyu et al., 2023), counterfactual (Gat et al., 2023). The above methods alleviate the unfaithful issue either in a post-hoc manner or via costly training. We instead propose a inference-time method, which can improve both faithful and accurate for different reasoning tasks, also maintain a similar computation cost as beam search.

## 3 PROBABILISTIC INFERENCE FOR FAITHFUL RATIONALE GENERATION

We firstly introduce the faithfulness definition in our context, which serves as a foundation for our algorithm design (§3.2). Then, we frame the faithfulness-controllable generation problem as a form of posterior inference under constraints (§3.3). Finally, we elaborate on how our proposal distribution incorporates both local and global rewards to enhance faithfulness throughout the entire reasoning process (§3.4).

### 3.1 PROBLEM DEFINITION

As our framework is grounded in a Monte Carlo search algorithm, we formalize the generation process as a search problem, represented as $\langle S, \mathcal{V}, \pi, U \rangle$. The state space $S$ consists of multi-token sequences drawn from the vocabulary $\mathcal{V}$. The transition function $\pi_t(x_{t+1} \mid x_t) \in \Delta^{|\mathcal{V}|}$ outputs a probability distribution over $\mathcal{V}$. The reward function $U$ guides the search process. The objective is for the model to reach a terminal state defined by $|eos|$ token, producing a sequence $y = \langle v, v', \ldots, |eos| \rangle$. Special emphasis is placed on identifying a faithful rationale and achieving accurate answer prediction, guided by $U$.

### 3.2 FAITHFUL RATIONALE GENERATION

Faithfulness is a crucial aspect of interpretability, and there are many different definitions and evaluation schema based on perturbation (Alvarez Melis & Jaakkola, 2018). Specially, we calculate the prediction difference before and after adding or removing important tokens from the input, i.e., $f(X) - f(X/I)$, where $X$ is input for the model $f(\cdot)$ and $I$ represents an important span within the input. This evaluation can be adapts to continuous changes, such as logits shifts and rationale changes (Siegel et al., 2024).

In our settings of LLM reasoning tasks, faithfulness applies to both the generated answers (such as classification labels in text classification tasks) and the rationales:

- The generated rationales should accurately reflect the true reasoning process, which can be assessed through perturbation.
- The label and rationale should be consistent with each other, i.e., the rationale should correctly reflect the predicted label.

In a broad sense, a faithful model would pay attention to the context provided (both the user query and the generated label) and make corresponding changes in their responses when facing with input context alterations. However, empirical observations (Lin et al., 2024) show that LLMs, even those instruction-tuned ones, tend to generate similar token distributions across most output positions. Inspired by the successes of injecting domain knowledge into LLMs reasoning (Ge et al., 2024), it could be promising to introduce domain-expert knowledge to make the model more sensitive and responsive to the given context.

### 3.3 PROBABILISTIC INFERENCE FOR RATIONALE GENERATION

During LLM inference, the transition distribution (Markov chain) $\pi$ is used to sample the next token given a prefix string according to the model's pretrained distribution. Existing research shows that LLMs are inferior in generating accurate answers and faithful rationales for out-of-distribution inputs, sometimes performing even worse than pretrained smaller expert models (Yuan et al., 2023; Gekhman et al., 2024). Therefore, our goal here is to introduce reward from expert models to adjust the LLMs' output distribution.

**Background of Feynman-Kac model.** Feynman-Kac formulae (Del Moral & Del Moral, 2004) is designed to admit probabilistic sequential Monte Carlo approximation (Lew et al., 2023), which involves a tuple consisting of an initial state, a transition distribution, and a potential function $(s_0, \pi_t, G_t)$. In the context of generating tokens $s_t$ using model $f_\theta$, The potential function $G_t$ maps $(s_t, s_{t+1})$ to a non-negative score, analogous to the reward function. The adjusted probability of $f_\theta$ generates $s_t$ is calculated as follows:

$$\mathbb{P}_t(s_t) = \frac{\mathbb{E}_\pi \left[ \prod_{i=1}^{t \wedge T} G_i(S_{i-1}, S_i, f_\theta) \cdot [S_t = s_t] \right]}{\mathbb{E}_\pi \left[ \prod_{i=1}^{t \wedge T} G_i(S_{i-1}, S_i, f_\theta) \right]}, \tag{1}$$

where $[S_t = s_t]$ is an indicator function that is equal to 1 if the state at $t$ is $s_t$, and 0 otherwise. The numerator inside the expectation represents the product of rewards and the probability of reaching state $s_t$, ensuring that paths leading to high rewards over time are given more weight. Generation continues until a terminal token or the maximum length of the sequence $T$, i.e., $t \wedge T = \min(t, T)$.

**Probabilistic inference for faithful rationale generation.** We develop our faithfulness-seeking model based on the Feynman-Kac framework primarily for its *computation efficiency* and *lookahead rewards*. Unlike existing *explicit* MCTS requiring expensive rollouts or simulations to evaluate potential actions (Xie et al., 2024; Zhang et al., 2024a; openai, 2024), it computes expected rewards in a more integrated and efficient way, streamlining the inference process. The incorporated lookahead $G(s_{t-1}, s_t)$ is achieved on cumulative rewards across multiple steps , rather than overly prioritize short-term gains or rely on heuristics, e.g., the normalized average (Wang et al., 2024) that do not model future states effectively.

In our settings, the generated sequence for reasoning tasks consists of an answer and an explanation [2]. The design of the faithfulness-related potential function takes into account (i) the accuracy of the predicted answer and (ii) the faithful rationale that explains the prediction. Since the answer is generated first without considering the subsequent context, whereas a faithful rationale requires coherence with the surrounding context, we introduce a local expert and a lookahead expert to address these two aspects. The generation framework is outlined in Algorithm 1: Dual-Reward Probabilistic Inference, with functions LOCALMASK and GLOBALREWARD.

At each timestep during model inference, both the local and global experts calculate a weight, denoted as $\alpha_t$, for the generated token $x_t$. Similar to beam search, we generate $K$ branches and select the generated trajectory with the highest sequence-level weight. For the local reward, we adjust the token's probability to favor predictions that align with the local expert's output $c_0$. For the global reward, which encourages rational generation, we use the global expert model $g$ to score the current token $x_t$ based on future predictions.

### 3.4 SEARCH WITH DUAL-REWARD

**Local mask.** One heuristic and lightweight approach to constrained generation from LLMs is to use masking or logit bias to reweigh the probabilities of sampled tokens, i.e., $\pi_t$. Many existing methods (Liu et al., 2024a; Zhao et al., 2024) leverage generation logits from smaller models to calibrate the logits from larger

---

[2]To prevent scenarios where an overly long rationale causes the answer to exceed the output length limit, we prioritize generating the answer first. Our framework can be extended to tasks where the answer space is infinite (Appendix C.1).

---

**Algorithm 1** Dual-Reward Probabilistic Inference

---

**Input:** model $f_\theta$, state transition function $\pi_t$, beam size $K$, max length $T$, label set $\mathcal{C}$, vocabulary $\mathcal{V}$
    terminal token $|eos|$, initialised weighted input sequence $\{(x_t, \alpha_t) \leftarrow (s_0, 1)\}_{t=1}^{T}$.
**Input:** Local expert prediction answer $c_0$, global expert transition function $U^g$.
**Output:** selected generated token sequence $x_t^{k^*}$, $t = 1, ..., T$

    **While** $t < T$ and $x_t \neq |eos|$ **do**
        **if** t == 0 **then**                      ▷ First position for answer generation
             $x_{t+1}^{k} = \text{LOCALMASK}(\pi_t, x_t, f_\theta, C)$ for $k = 1, ..., K$
        **else**                         ▷ Other positions for rational generation
             $x_{t+1}^{k} \sim \pi_t(x_{t+1}^{k} | x_t^{k}, f_\theta)$ for $k = 1, ..., K$
        **end if**
        $\alpha_{t+1}^{k} = \text{GLOBALREWARD}(\pi_t, x_{t+1}, f_\theta, U^g)$ for $k = 1, ..., K$
    **end while**
    $k^* = \arg\max_k \left( \sum_{t=1}^{T} \alpha_t^k / T \right)$   ▷ Select the $k$-th sequence with maximum sequence-level weight

    **function** LOCALMASK$(\pi_t, x_t, f_\theta, \mathcal{C}, c_0)$
        $\mathcal{V}' \leftarrow \mathcal{V} \setminus (\mathcal{C} \setminus \{c_0\})$                ▷ Remove the tokens in set $\mathcal{C}$ (except $c_0$) from $\mathcal{V}$
        $\pi_t'(x_t = v' \mid x_{t-1}) = \frac{\pi_t(x_{t+1}=v'|x_t)}{\sum_{v' \in \mathcal{V}'} \pi_t(x_{t+1}=v'|x_t)}$          ▷ Reweigh the token distribution $\pi_t$
        $v \sim \pi_t'$
        **return** $v$

    **function** GLOBALREWARD$(\pi_t, x_t, f_\theta, U^g)$
        $v \sim \pi_t(x_{t+1} | x_t, f_\theta)$
        $\alpha_{t+1} \leftarrow U^g(x_t, v)$                 ▷ Score the $x_t$ using the global expert model on $(x_t, v)$
        **return** $\alpha_{t+1}$

---

model, e.g., logit fusion, in order to alleviate undesirable attributes such as toxicity and untruthful. However, these attributes are implicitly conveyed over longer spans rather than individual tokens, making token-level constraints insufficient (See the performance of logit fusion in Table 7). Therefore, we don't adopt such local constraint for faithfulness enhancement. Instead, domain-specific experts tend to demonstrate better accuracy in knowledge-rich tasks. Specially, we introduce a set of classification label words $\mathcal{C}$ and we remove the label words which are not included in the expert model's prediction $c_0$. We then renormalise the output probability based on the new vocabulary set $\mathcal{V}'$ (Details in Function LOCALMASK) in Algorithm 1.

**Global reward.** As mentioned earlier, the local hard constraints may fail in preventing undesirable attributes; therefore, we require global and lookahead constraints. Similar to the rollout phase in Monte Carlo search, we allow the generation of multiple promising trajectories and select the optimal one based on the overall rewards. At each step, the exploited reward is defined by the hindsight function $U_t^g$ defined by global expert. Specifically, we use the global expert model to score the generated n-gram $(x_t, x_{t+1})$ from $f_\theta$ (Details in function GLOBALREWARD). It is expected that text spans faithful and coherent with the context are preferred, as they better align with the expert's domain-specific distribution (The effects of global reward in faithfulness enhancement are summarized in Table 6).

## 4 TASK PERFORMANCE AND FAITHFULNESS EVALUATION

We conduct experiments on three tasks: *student answer assessment*, i.e., ASAP [3], *natural Language Inference (NLI)*, including the Stanford Natural Language Inference (*SNLI*) (Bowman et al., 2015) and the Multi-

---

[3] https://kaggle.com/competitions/asap-sas

Genre Natural Language Inference (MNLI) (Williams et al., 2018) datasets; and *TruthfulQA* dataset (Lin et al., 2022). In each of these tasks, LLMs are required to generate both class labels and rationales justifying their classification decisions. For student answer assessment, the labels represent valid score ranges, 0-5; For NLI, the labels are '*entailment*', '*contradiction*', or '*neutral*'. For *TruthfulQA*, we use a subset of the dataset converted into a multiple-choice format.

**Backbone models and expert models.** Our study employs two widely used instruction-tuned LLMs as our backbone models: Llama-3-8B-Instruct (Dubey et al., 2024) and Mistral-7B-Instruct-v0.3 (Jiang et al., 2023)[4]. For each dataset, we incorporate one expert model for local reward and another expert model to offer global reward. The details are shown in Appendix A. Note that all the expert models are only fine-tuned on the validation subset. It is worth noting that our framework can integrate with various expert models, even when their tokenization spaces differ from the backbone models.

## 4.1 MAIN RESULTS OF TASK PERFORMANCE

**Our method significantly outperforms the backbone model, achieving a substantial performance improvement.** The experimental results summarized in Table 2 highlight the efficacy of our decoding method, which combines both local and global rewards, compared to the baseline Llama3 across various NLP tasks. Due to limited computational resources, our experiments are evaluated on 100 randomly sampled instances from each dataset, with the model utilizing 8-bit quantization. Results show the consistent enhancement in task performance achieved by our method without any further tuning of the backbone model on task-specific datasets. Specially, our reward-based decoding strategy achieves an average 40% improvement in accuracy on *Student Answer Scoring*, a 20% improvement on *NLI*, and a 26% increment on *TruthfulQA*.

## 4.2 RATIONALE FAITHFULNESS EVALUATION

**Perturbation.** Following existing approaches for counterfactual generation in faithfulness evaluation, we modify key parts $w$ of the inputs $I$ and observe the resultant variations in the generated rationales. For *student answer assessment*, we remove the clause (sub-sentence) from the student answer that is most semantically related to the original rationale $R_o$. In the case of *NLI* and *TruthfulQA*, where the sentences in the provided context are typically very short (often a single sentence), we introduce perturbations through word insertion, as inspired by Atanasova et al. (2023). Specifically, we apply POS tagging to the sentence in the context to identify verbs and adjectives which are likely to have a greater impact and replace them with alternative words. We then feed the perturbed input to the model for new rationale $R_n$. The algorithm for generating counterfactual rationales is detailed in the Appendix A.

| Datasets | Llama3 | Ours |
|---|---|---|
| *Student Answer Assessment* | | |
| ASAP-Q1 | 28% | **57%** |
| ASAP-Q2 | 28% | **68%** |
| ASAP-Q3 | 45% | **90%** |
| ASAP-Q4 | 38% | **84%** |
| *Natural Language Inference* | | |
| SNLI | 49% | **69%** |
| MNLI | 57% | **77%** |
| *Question-Answering* | | |
| TruthfulQA | 47% | **73%** |

Table 2: Task performances (Accuracy) across three tasks between Llama3 and our proposed method.

**Evaluation Metrics.** For the sub-sentence removal perturbation, we calculate the semantic relatedness between the removed text span $w$ and the original rationale, denoted as $S_{wo} = \text{Sim}(w, R_o)$, and between the removed span and the new rationale, denoted as $S_{wn} = \text{Sim}(w, R_n)$. A faithful model is expected to produce a significant **semantic variation**, i.e., $\Delta(S_{wo} - S_{wn})$, as the removed sub-sentence should be closely related to the original rationale but less similar to the new rationale. For the *word insertion* perturbation, we calculate the percentage of new rationales that include the newly inserted word, denoted as **word inclusion**. Both large semantic variation and word inclusion indicates a better faithfulness.

---

[4]The results of Mistral is presented in Table 8 in Appendix.

### 4.2.1 RESULTS OF FAITHFULNESS EVALUATION

**Results on *student answer assessment*.** Table 3 compares the semantic variations $\Delta(S_{wo} - S_{wn})$ measured in ROUGE scores (R-1,R-2 and R-L).We expect that We can observe in Table 3 that the variation in ROUGE scores across different subsets demonstrates the effectiveness of our method in generating more faithful rationales compared to the baseline model. Our method shows significantly enlarged ROUGE-L score differences for all the subsets except for Q2, indicating a substantial lexical difference with the removed content after applying counterfactual modifications. For Q4, the notable increase in all ROUGE metrics highlights a pronounced enhancement in the model's ability to retain relevant information despite the removal of critical sub-sentences.

| Method | ASAP-Q1 | | | ASAP-Q2 | | | ASAP-Q3 | | | ASAP-Q4 | | |
|---|---|---|---|---|---|---|---|---|---|---|---|---|
| | R-1 | R-2 | R-L | R-1 | R-2 | R-L | R-1 | R-2 | R-L | R-1 | R-2 | R-L |
| Llama3 | 0.037 | **0.043** | 0.034 | 0.052 | **0.050** | **0.052** | 0.042 | 0.031 | 0.043 | 0.006 | 0.018 | 0.001 |
| Ours | **0.064** | 0.035 | **0.052** | **0.056** | 0.029 | 0.050 | **0.064** | **0.051** | **0.058** | **0.105** | **0.085** | **0.102** |

Table 3: The faithfulness on student answer scoring dataset, i.e., sentence-level *semantic variations* measured in ROUGE scores.

| Dataset | Llama3 | Ours |
|---|---|---|
| SNLI | 11% | **15%** |
| MNLI | 9% | **18%** |
| TruthfulQA | 2% | **18%** |

Table 4: Faithfulness on *NLI* and *TruthfulQA* tasks on *word inclusive*.

**Results for *NLI* and *QA*.** The *word inclusion* metric in Table 4 enriches our analysis by quantifying the extent to which perturbed input tokens are reflected in the generated rationales. For the *SNLI*, word inclusion increases by 4%, and for the *MNLI* dataset, it raises by 9%. These increases suggest that our model not only modifies its responses but does so in a way that better captures the new input elements. For *TruthfulQA*, our improvements are significantly more pronounced, with a 16% enhancement in performance. These higher inclusion rates highlight the model's capacity to maintain high fidelity to the input semantic changes.

## 5 MODEL ABLATION AND CASE STUDY

| Datasets | Expert Model | | Our + Expert Model | | |
|---|---|---|---|---|---|
| (Backbone) | CLS | Expert | Ours+CLS | Ours+Expert | Our (Full) |
| *Student Answer Assessment* | | | | | |
| Q1 (28%) | 85% | **76%** | 55% | 45% | 57% ↑ |
| Q2 (28%) | 72% | 48% | 51% | 52% | **68%** ↑ |
| Q3 (45%) | 91% | 71% | 64% | 79% | **90%** ↑ |
| Q4 (38%) | 88% | 67% | 66% | 71% | **84%** ↑ |
| *NLI* | | | | | |
| SNLI (49%) | 86% | **76%** | 54% | 54% | 69% ↑ |
| MNLI (57%) | 88% | 76% | 41% | 61% | **77%** ↑ |
| *TruthfulQA* | | | | | |
| TruthQA (47%) | 100% | 70% | 30% | 46% | **73%** ↑ |

Table 5: Ablation performance across three datasets in **accuracy**. ↑ denotes better than backbone.

| Dataset | Expert | Our+Expert | |
|---|---|---|---|
| (Backbone) | Expert model | Our+Expert | Our(full) |
| *Student Answer Assessment* (semantic variation ↑) | | | |
| Q1 (0.034) | **0.094** | 0.032 | 0.052 ↑ |
| Q2 (0.051) | **0.114** | 0.048 | 0.050 |
| Q3 (0.042) | **0.061** | 0.037 | 0.058 ↑ |
| Q4 (0.001) | 0.085 | 0.053 ↑ | **0.102** ↑ |
| *NLI* (Word inclusive ↑) | | | |
| SNLI (11%) | 13% | **22%** ↑ | 15% ↑ |
| MNLI (9%) | 9% | **19%** ↑ | 19% ↑ |
| *QA* (Word inclusive ↑) | | | |
| TruthfulQA(2%) | **24%** | 19% ↑ | 18% ↑ |

Table 6: Ablation performance across three datasets in **faithfulness**. ↑ denotes better than backbone.

### 5.1 EFFECTS OF LOCAL AND GLOBAL REWARD

We show the results of ablating the local reward, i.e., classifier (CLS) and global reward, i.e., Expert for task performance and faithfulness in Table 5 and Table 6.

**Local reward contributes to task performance.** We observe significant performance improvements by introducing the local reward over *student answer assessment* and *SNLI*. The increment is due to the expertise of local reward in classification. Notably, when incorporating the CLS, our method does not necessarily perform as well as the classifier alone. This is partly because we could only penalize the words in label space but not all synonyms to the label word. For example, in *MNLI*, the model insists on *'polarization'* even we lower the probability of *'contradictory'*.

**Global reward contributes to both task performance and faithfulness.** From Table 5, with the incorporation of global expert, we achieve a better task performance than backbone on all datasets expect *TruthfulQA*. Some improvements are even higher than CLS, especially on the *Q2-Q4*, *MNLI*. It can be explained that the task performance can also benefit from domain knowledge, which is consistent with the observations in Ge et al. (2024). From Table 6, Expert enhances the model faithfulness across the three benchmarks, although slightly lower on Q1-Q3, its improvements on Q4 is much more significant, from 0.001 to 0.053.

**Local reward fails to give look-ahead control by comparing with *logitfusion* (Liu et al., 2024a).** It is expected that our proposed local constraint is mainly for providing accurate label information, rather than rationale generation. To highlight the advantages of our global reward, we compare with a decoding method with local constraint (Liu et al., 2024a), which fuses the generated token probability from the expert and the backbone models at each timestep via interpolation. As shown in Table 7, rationales generated from logits fusion baseline are among 55% less faithful on average compared with ours (displayed in Table 3) [5], their task performance is even lower than some of the backbone results. Moreover, logit fusion does not support fusion between models with different tokenization. The theoretical advantage between logit fusion and our look-ahead reward is that our method *considers future tokens' plausibility when scoring the currently generated token*. The potential function $U^g$ at timestep $t$ considers the generated sequence until $l$ steps ahead.

| Datasets | Acc | Faithfulness |
|---|---|---|
| Q1 | 34% (-40%) | 0.028 (-46%) |
| Q2 | 22% (-68%) | 0.037 (-26%) |
| Q3 | 40% (-56%) | 0.019 (-67%) |
| Q4 | 29% (-65%) | 0.019 (-81%) |
| Avg | 31% (-57%) | 0.026 (-55%) |

Table 7: Logit fusion results on both task performance and faithfulness for student essay assessment. The relative changes compared with ours are in the bracket.

**Local and global rewards jointly achieves the overall best task performance and faithfulness.** Interestingly, our full model outperforms the inclusive of either CLS or Expert alone, achieving the best task performance across all seven datasets and demonstrating greater faithfulness than Our+GExpert in five of them. This joint effect can be inspired by pruned Monte Carlo search, where undesirable branches are eliminated. Similarly, our local constraint serves the same purpose by removing trajectories that lead to undesirable label words. More strictly, combined with the stochastic nature of the decoding process, we give a proof that branch pruning can lead to different search trajectories in Appendix D. This explains why our full model achieves overall better performance than one that solely incorporates the Expert model.

## 5.2 DOMAIN-SPECIFIC WORD DISTRIBUTION

We utilize TF-IDF to select domain-specific words (after removing the stopwords) from the student responses in the *student answer assessment* dataset. The selected words and their associated TF-IDF scores are depicted in the blue curve (context) in Figure 2. Since the TF-IDF score reflects the importance of these contextual words, we calculate the TF-IDF scores for the same words within the rationales generated by the backbone model (in orange) and our-full model (in green). This allows us to verify whether the generated rationales align well with the important spans in the context. It is clear that the green curve is mostly above the orange curve,

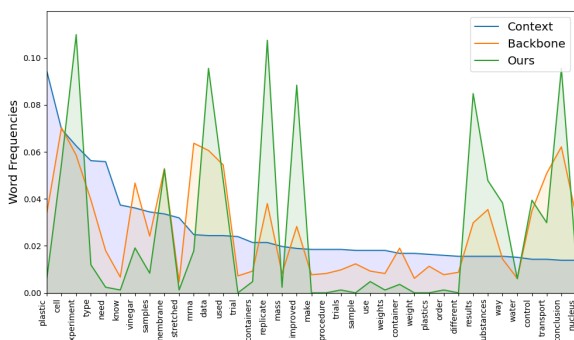

Figure 2: Distribution of domain-specific word.

---

[5]Note that more than 50% of the samples failed to generate responses due to the fused logit being impractical for the backbone model.

showing that our method can respond more actively to those domain-specific words, such as "*experiment*", "*data*", "*replicate*", "*substances*", and "*nuclear*".

## 5.3 CASE STUDIES

To highlight the differences between the rationales from the backbone model and ours, we randomly select two examples from *Student Answer Assessment* and *Natural Language Inference* datasets, as shown in case study 5.3. For the student answer assessment task, there are three key elements (highlighted in colour) mentioned in the context, such as *how much vinegar*, *what size surface*. The student response also mentioned these key concepts. In the rationale generated by the backbone, its feedback is not aligned well with the key concepts, like *'fails to specify the vinegar type, drying method'*. Instead, our generated rationale closely related to the two elements (spans are highlighted in colours).

For the *NLI* dataset, we present a perturbation example where the word *frugally* is inserted at the beginning of the context. Our method-generated rationale could capture this word sensitively and be reflected in the rationale content. While the backbone model acts less robust, it completely ignores this new insertion and incorrectly predicts the label.

---

**Case studies**

### Student Answer Assessment

**Key Answer Elements**: (i) how much vinegar was used in each container. (ii) what size/surface area of materials should be used. (iii) what drying method to use. . . .

**Student Response**: "Some example of things you would need to no is how much vinegar is being poured in each of the separate containers. Another thing is the volume of each substance because the surface area will effect how much is eroded off. Another thing where you let the samples sit for 30 minutes because there might be difference in temperature effecting how much it drys."

* **Backbone:** Score: 1 point; The student has provided a valid concern about the need for specific information to replicate the experiment, but only mentions three key details that would be necessary to know, *and **fails** to specify the type of vinegar used, drying method*, or other important details.

* **Ours**: 2 points; The student's response describes two additional pieces of information that would be needed to accurately replicate the experiment,the amount of vinegar used and the size/surface area of the materials.

### Natural Language Inference

**Perturbed Premise**: frugally Requires free registration.

**Hypothesis**: Does not require free registration.

* **Backbone:** Entailment; Requires free registration is a necessary condition for only if Requires free registration.

* **Ours:** Contradiction; The premise states that the website [frugally] requires free registration, which implies that a user must provide some information or sign.

---

## 6 CONCLUSION

In this paper, we address the critical challenge of enhancing both accuracy and faithfulness in large language models. Specially, we introduce a probabilistic inference paradigm that incorporates fine-grained and look-ahead rewards to search desirable trajectories. Compared to existing inference-time solutions, our method distinguishes itself through a domain-specific proposal distribution that increases the model's responsiveness to key words in the context. We verify the effectiveness of our approach across three diverse datasets, evaluating both task performance and faithfulness metrics. Furthermore, our model ablation study demonstrates the superiority of integrating both local and global rewards.

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

APPENDIX

# A  EXPERIMENT SETUP

We utilize a comprehensive experiment approach with seven datasets across three distinct reasoning tasks. The backbone models adopted for experiments, dataset details, counterfactual generation algorithms and hyper-parameters setting are outlined below.

**Backbone model choice.**    Our experiments select two commonly used backbone model choices, Llama3-8B Instruct model[6] and Mistral 7B Instruct model[7] models. Both models are downloaded from the Hugging-Face Models' space, and we used the Huggingface Transformer [8] to implement the models.

**Hyper-parameters for inference.**    For efficient model inference, we applied 8-bit quantization on both the backbone and global reward models for *Our+Expert* and *Our(full)* experiments. The local reward models are loaded without any quantization. We set a maximum allowance of 10 different particles during decoding, which means it will keep a maximum of 10 different paths during the search through different weighted decoding paths. The beam factor for expanding searching at each particle is set as 3. To optimize the computational resources for generation, we applied different maximum token length sizes for each task, which we will introduce under each task. Also, our local rewards will be disabled once the answer token is generated to remove the token space constraint. We use a batch size of 64 to inference our framework on a single NVIDIA A100 40G graphic card. The random seed has been set as 42 for all the components.

**Predicted label evaluation details.**    Apart from the faithfulness evaluation details presented in Section 4, the evaluation for the predicted label is extracted and compared with the ground-truth label to calculate the accuracy score. Following each prompt template, we designed a regular expression to extract the score/labels from the generated sequence. If the model fails to follow the prompt to generate a valid label token, then it counts as a wrongly predicted instance. In short, only correctly predicted instances that follow the prompt required output pattern count towards the accuracy score.

## A.1  STUDENT ANSWER ASSESSMENT SETUP

We employed the ASAP [9] dataset to evaluate our methods' effectiveness. Following the rationale generation paradigm established by Li et al. (2023), we adopted the same rationale generation prompt used in their study, focusing on four subsets of science and biology questions. For each dataset, we randomly selected 100 instances from the test split. All the local and global reward models are trained on a training set without data contamination. Our analysis shows that previous zero-shot students' answer assessment rationale generation method typically generates rationales with an average sequence length of less than a hundred. Therefore, we set the maximum token length for this task as 100.

**Prompt template.**    We use the following prompt template applied to all our test instances. The question, key_elements, marking_rubric, and student_answer correspond to question-dependent information from the dataset:

---

[6]https://huggingface.co/meta-llama/Llama-3.1-8B-Instruct
[7]https://huggingface.co/mistralai/Mistral-7B-Instruct-v0.3
[8]https://huggingface.co/docs/transformers/index
[9]https://kaggle.com/competitions/asap-sas

> **[Question]:** {*question*}
> **[Key Elements]:** {*key_elements*}
> **[Marking Rubric]:** {*marking_rubric*}
> **[Student Answer]:** {*student_answer*}
> Please assess this student response and provide rationale, in the format of "x point/s; rationale":

**Local and global reward model setup.** We utilize a text classifier fine-tuned on the ASAP datasets, built on DeBERTa-v3-large model (He et al., 2023) as the local reward model. We adopted two global model choices: **Choice 1**: An open-source explainable student answer scoring LLM developed by Li et al. (2024a). The model is fine-tuned using synthetically generated student answer data with 4-bit quantization with LoRA. **Choice 2**: A mistral 7B model fine-tuned on science question and answer: Weyaxi/Einstein-v2-7B. The model is fine-tuned with a science question-and-answer dataset based on the Mistral 7B model.

**Sentence-level perturbation for *student answer assessment* dataset.** Our evaluation strategy involved systematically modifying key parts from student answers $x_i$ in the input data by comparing each key answer element $k_i$ from all key elements $K$. Then, observe the resultant variations in the generated rationales. By doing so, we could ascertain whether the rationales remained consistent and aligned with the altered inputs, thereby providing insights into their reliability and interpretability. This approach ensures that the rationales are contextually relevant and robust against variations in input, thereby enhancing their practical utility in real-world applications.

---

**Algorithm 2** Student Answer Perturbation Algorithm

---

1: **procedure** PERTURBATION($x_i, K$)
2:     $\mathbf{S} \leftarrow$ Tokenize($x_i$)
3:     $\mathbf{I} \leftarrow$ array of zeros with length($|\mathbf{S}|$)
4:     **for** $j \leftarrow 1$ to $|\mathbf{S}|$ **do**
5:         **for** $k \leftarrow 1$ to $|K|$ **do**
6:             $\mathbf{I}[j] \leftarrow \mathbf{I}[j] + \text{Sim}(\mathbf{S}[j], K[k])$
7:         **end for**
8:     **end for**
9:     $i_{\max} \leftarrow \text{argmax}(\mathbf{I})$
10:     $\mathbf{S} \leftarrow \mathbf{S} \setminus \{\mathbf{S}[i_{\max}]\}$
11:     $\hat{\mathbf{S}} \leftarrow$ Join($\mathbf{S}$)
12:     **return** $\hat{\mathbf{S}}$
13: **end procedure**

---

## A.2 NATURAL LANGUAGE INFERENCE (NLI)

For NLI, we utilized two key datasets: the Stanford Natural Language Inference (*SNLI*) (Bowman et al., 2015) and the Multi-Genre Natural Language Inference (*MNLI*) (Williams et al., 2018) datasets. These datasets are critical for assessing the ability of our models to handle a range of inferential relationships across various genres, thus providing a comprehensive view of model performance in understanding language context. We randomly selected 100 instances from the official validation split for each dataset; for the MNLI dataset, we used the matched validation set. We find existing explanations from the ESNLI dataset have an average sequence length shorter than 30 tokens. Therefore, we employed the maximum token length of 30 for the NLI task.

**Prompt template.**    We use the following prompt template to evaluate our method on all the NLI tasks. The premise and hypothesis placeholder corresponds to the premise and hypothesis from each row.

> **Here is a premise:** {*premise*}
> **Here is a hypothesis:** {*hypothesis*}
> Please choose whether the hypothesis is entailment, neutral, or contradiction to the premise, and provide a rationale for your choice. Output the label and rationale in the format of "Prediction: [label]; [explanation]":
> Prediction:

**Local and global reward model setup.**    We use an open-source fine-tuned BART model (Lewis et al., 2020) for performing classification in NLI tasks as the local reward model. Please refer to their released repository for detailed training data usage and splits. For the global model, we utilize a LoRA fine-tuned Llama-2-7B model on the ESNLI dataset Camburu et al. (2018). The model is trained solely on the training set of the ESNLI. To reduce computational resources, the local reward model is disabled after generating the answer token.

**Hyper-parameters for inference.**    For efficient model inference, we applied 8-bit quantization on both the backbone and global reward models. The local model is loaded without quantization. We set a maximum allowance of 10 different particles during decoding. The beam factor for searching is set as 3, with a maximum token length of 30.

**Word-level perturbation for *NLI* tasks.**    For *NLI*, we identify a keyword among adjective and verb words by POS-tagging using *TokenizeAndTag*. The adj and adv word lists are imported from the nltk package. Once the words are tagged, we randomly insert an irrelevant adjective word into either the premise or hypothesis to create such perturbation using the GenerateExample function. The GenerateExample function takes the whole token lists and the randomTarget word and position to reconstruct the perturbed sequence. Our goal is to detect the modified word from the generated rationale to evaluate the faithfulness of our method.

---

**Algorithm 3** NLI Word Perturbation Generation

---

1: **procedure** PERTURBATION($x_i$, adj, adv)
2:     tokens, tags ← TokenizeAndTag($x_i$)
3:     targets ← IdentifyTargets(tags, adj, adv)
4:     randomTarget ← SampleTargets(targets)
5:     example ← GenerateExample(tokens, randomTarget)
6:     **return** example
7: **end procedure**

---

## A.3    QA

The *TruthfulQA* dataset contains questions and answers. Each question has multiple answers, which were adapted into a multiple-choice format. The model's task for this dataset is to select the most truthful answer through all the options.

**Prompt template.**    We use the following prompt template to evaluate our method on the QA task. The question is the question row from the dataset, and the choices are selections for answers from the dataset.

> **Question:**  {*question*}
> **Choose the best answer from following options:** {*choices*}
> Output the selection with reason in the format of Answer: "choice; reason". Answer:

**Local and global reward model setup.** We use an open-source truth judge released by Allen AI: allenai/truthfulqa-truth-judge-llama2-7B. For the global model, we utilize a 7B LLM specialized in truthful QA, released by Zhang et al. (2024b). Please refer to the original paper for the detailed training setup and dataset split for the local and global models. To reduce computational resources, the local reward model is disabled after generating the answer token.

**Hyper-parameters for inference.** For efficient model inference, we applied 8-bit quantization on both the backbone and global reward models. The local model is loaded without quantization. We set a maximum allowance of 10 different particles during decoding. The beam factor for searching is set as 3, with a maximum token length of 30.

**Word-level perturbation for *QA* task.** For *QA* task, we identify an influential word to be replaced, similar to the Algorithm 3. Instead of using an algorithm to perturb the word, in this task, we query the GPT-4 model to modify the original sentence and output both the modified word and the perturbed sentence. Evaluating the faithfulness of the task still depends on the successful rate of reflection of modified words from the rationale.

## B ADDITIONAL EXPERIMENT RESULTS

### B.1 EFFECTS OF DIFFERENT BACKBONE MODEL

**Similar experimental result trends are also observed on other backbone models.** In addition to evaluating our method on the Llama3-8B model, we extended our investigation to other backbone models, specifically examining the Mistral 7B model. The results, summarized in Table 8, demonstrate a consistent trend across different model sizes, underscoring the robustness of our method.

The performance comparison conducted on the Mistral 7B model reveals an overall slight decrease in average performance compared to the Llama3-8B results. Despite this reduction, the relative performance enhancements provided by our method remained consistent. For instance, while the baseline Backbone model scored lower across all tasks, introducing local and global expert knowledge notably improved performance, with our method achieving the highest task performance in all the cases.

Moreover, we also evaluate the faithfulness of the rationale for *Student Answer Assessment* in Table 9 and *NLI* and *TruthfulQA* in Table 10. Similar to our observation with Llama3-8B as the backbone model, the faithfulness evaluation results highlight that our method is able to generate a more faithful rationale that could reflect the change in the input compared with the backbone model.

| Datasets | Backbone | Ours |
|---|---|---|
| *Student Answer Assessment* | | |
| ASAP-Q1 | 31% | **80%** |
| ASAP-Q2 | 38% | **69%** |
| ASAP-Q3 | 35% | **84%** |
| ASAP-Q4 | 45% | **80%** |
| *NLI* | | |
| SNLI | 47% | **79%** |
| MNLI | 41% | **78%** |
| *QA* | | |
| TruthfulQA | 43% | **72%** |

Table 8: Mistral 7B model overall performance compared across each method. The best performance over text generation models is highlighted in **bold**.

These findings indicate that while the inherent capabilities of the backbone models influence absolute performance metrics, the efficacy of our approach in enhancing model output by integrating classifiers and expert insights transcends the specific model used. The trend observed with the Mistral-7B, similar to that with the LLama-3-8B, validates our method's general applicability and effectiveness across different neural architectures, highlighting its potential for broad adoption in diverse NLP tasks.

| Method | ASAP-Q1 | | | ASAP-Q2 | | | ASAP-Q3 | | | ASAP-Q4 | | |
|---|---|---|---|---|---|---|---|---|---|---|---|---|
| | R-1 | R-2 | R-L | R-1 | R-2 | R-L | R-1 | R-2 | R-L | R-1 | R-2 | R-L |
| Mistral | 0.005 | 0.004 | 0.005 | 0.024 | 0.006 | 0.023 | 0.026 | 0.015 | 0.028 | 0.012 | 0.003 | 0.005 |
| Ours | **0.046** | **0.033** | **0.042** | **0.080** | **0.078** | **0.080** | **0.032** | **0.019** | **0.034** | 0.010 | **0.012** | **0.008** |

Table 9: The sentence-level *semantic variations* measured in ROUGE scores.

| Dataset | Mistral | Ours |
|---|---|---|
| SNLI | 14% | **15%** |
| MNLI | 7% | **16%** |
| TruthfulQA | 17% | **20%** |

Table 10: Faithfulness on *NLI* and *TruthfulQA* tasks on *word inclusive*. Ours greatly enhances the faithfulness on both datasets.

## B.2 Investigate the Lexical Similarity

To explore the source of faithfulness enhancement, we calculate the semantics overlap between the given context and generated rationale. This is motivated by the hypothesis that a more faithful model will generate a rationale more coherent with the given question context rather than too general and nonsense words or hallucinations. Therefore, we calculate the BLEU between the generated rationale and the given prompt, including the question, student answer and instruction. Results are shown in Table 11.

| Method | 1-gram | 2-gram | 3-gram | 4-gram |
|---|---|---|---|---|
| Backbone | 0.106 | 0.090 | 0.058 | 0.025 |
| Ours | **0.452** | **0.333** | **0.167** | **0.058** |

Table 11: Semantic coherence between given assessment marking criteria and generated rationale. Higher values imply higher lexical similarity.

## C Generalisability of Our Generation framework

### C.1 Infinite label space

Our method is extendable to scenarios with an infinite label space ($|\mathcal{C}| = \infty$), even though the current evaluations are performed on tasks where the label space is constrained ($|\mathcal{C}| = N \in \mathbb{R}$). For instance, in mathematical problem-solving, the answer can be any arbitrary number. In such cases, the expert model provides a prediction $M$, with its confidence expressed as the probability $w_1$ assigned to $M$, and $w_2$ to the second most probable prediction. The ratio $\frac{w_1}{w_2}$ serves as an indicator of the expert's confidence in delivering $M$ (Moon et al., 2020). This confidence is then used as a multiplier to enhance the backbone model's prediction for $M$. Finally, the backbone model's transition distribution is renormalized to maintain a valid probability distribution.

### C.2 Expert Models Across Different Tokenisation Spaces

| | | ASAP-Q1 | | | ASAP-Q2 | | | ASAP-Q3 | | | ASAP-Q4 | | |
|---|---|---|---|---|---|---|---|---|---|---|---|---|---|
| | Method | R-1 | R-2 | R-L | R-1 | R-2 | R-L | R-1 | R-2 | R-L | R-1 | R-2 | R-L |
| | Llama3 | 0.037 | 0.043 | 0.034 | 0.052 | **0.050** | 0.052 | 0.042 | 0.031 | 0.043 | 0.006 | 0.018 | 0.001 |
| | Ours (w/ Expert 2) | **0.124** | **0.142** | **0.126** | **0.065** | 0.025 | **0.063** | **0.090** | **0.097** | **0.092** | **0.104** | **0.116** | **0.107** |

Table 12: The sentence-level semantic variations measured in ROUGE scores with the global expert choice 2 from the Appendix A on *Student Answer Assessment*.

**Different expert model for *student answer assessment***  As highlighted in Appendix A, the second expert model employs a distinct tokenization strategy, and the model is trained and tailored specifically for scientific

| Method | Time Cost |
|---|---|
| Backbone (Beam Search) | 88 mins |
| :+ Local | 100 mins |
| :+ Global | 103 mins |
| Ours (full) | 116 mins |

Table 14: Computation cost for different methods on *Student Answer Assessment* Q4.

questions and answers. To further validate our approach, a series of experiments were conducted, as detailed in Table 13, using the *Student Answer Assessment* datasets. These experiments demonstrate the compatibility of our method with domain-specific expert models. Despite a modest reduction in performance compared to the highly specialized domain-related expert model, our method consistently outperformed the Llama 3 Backbone across all datasets. Further, as presented in Table 12, we assessed the faithfulness of the rationale generated under the guidance of the second expert model. Our findings indicate that our method maintained a high level of faithfulness to the perturbations, notwithstanding the different tokenisation approaches, underscoring its robustness and adaptability in handling domain-specific challenges.

**Dealing with rewards from different tokenisation models**  In our approach, we address the challenge of integrating rewards derived from various tokenization models used by expert systems. Specifically, after the generation of each token, it is converted into token IDs according to the expert model's tokenization scheme. Subsequently, rewards are calculated based on samples drawn from the expert model. This method ensures that the generated tokens are consistently evaluated in the context of the expert's linguistic framework, and therefore generating meaningful predicted rewards.

| Datasets | Backbone | Ours |
|---|---|---|
| *Student Answer Assessment* | | |
| ASAP-Q1 | 28% | **59%** |
| ASAP-Q2 | 28% | **55%** |
| ASAP-Q3 | 45% | **67%** |
| ASAP-Q4 | 38% | **56%** |

Table 13: Llama-3-8B performance on *Student Answer Assessment* with different global expert choices. The best performance over text generation models is highlighted in **bold**.

### C.3 COMPUTATION COST ANALYSIS

Although global and local constraints introduced new computations during the text generation processes, we didn't observe a huge computational cost increment in our method. As shown in Figure 14, we calculated the inference time on the Student Answer Assessment question #4 to compare the time used between methods on the same GPU. We use a beam size of 3 and a maximum of 100 tokens in generation settings. Compared with the backbone model, our method only increased by 32% on inference time. Compared to other sequential Monte Carlo method, such as *PPO-MCTS* (Liu et al., 2024b), which has a $2S$ times overhead compared to standard decoding from PPO models ($S$ is the number of simulations), our inference-time decoding maintains both the performance and greatly improve the computation efficiency.

## D PROOF OF PRUNED MONTE CARLO SEARCH

**Definition.**  We first define the notations: $\mathcal{A}, \mathcal{B}, \mathcal{C}$ are three searched trajectories, among which one trajectory will be pruned. $N_A, N_B, N_C$ are the number of simulations conducted on the corresponding trajectories, $W_A, W_B \ W_C$ are the total wins for the trajectories.

The estimated value of each branch, i.e., **the probability of being sampled** is defined as:

$$V_A = \frac{W_A}{N_A}, \quad V_B = \frac{W_B}{N_B}$$

Without loss of generalisability, we assume the initial condition and branch $C$ be identified and pruned:

$$V_A > V_B \implies \frac{W_A}{N_A} > \frac{W_B}{N_B}$$

Our proof goal is to show after pruning $\mathcal{C}$, the probability of sampling $\mathcal{B}$ can be larger than $\mathcal{A}$.

*Proof.* After pruning $\mathcal{C}$, the remaining resources (i.e., simulations) are redistributed to branches $A$ and $B$. We define $R_A$ and $R_B$ are the additional simulations allocated to $\mathcal{A}$ and $\mathcal{B}$.

After pruning, the new number of simulations for branches $A$ and $B$ are:

$$N'_A = N_A + R_A, \quad N'_B = N_B + R_B$$

After pruning: we define $W'_A, W'_B$ as new total wins after additional simulations. Therefore, the new values for $\mathcal{A}$ and $\mathcal{B}$ are as follows:

$$V'_A = \frac{W'_A}{N_A + R_A} \quad \text{(new estimated value of A)}$$

$$V'_B = \frac{W'_B}{N_B + R_B} \quad \text{(new estimated value of B)}$$

To establish that $V'_B > V'_A$, we require:

$$\frac{W'_B}{N_B + R_B} > \frac{W'_A}{N_A + R_A}$$

Cross-multiplying gives:

$$W'_B \cdot (N_A + R_A) > W'_A \cdot (N_B + R_B)$$

Given that $W'_B > W_B$ and $W'_A < W_A$, it is possible for the following to hold true. For example, in our *NLI* dataset, the undesirable labels are '*contradictory*', so we remove the trajectory $\mathcal{C}$ consisting of '*contradictory*'. For the remaining trajectories, $\mathcal{A}$ and $\mathcal{B}$ are related to '*contradictory*' and '*Neutral*' (not exact label, but similar attitude), respectively. With the removal of '*contradictory*', the new sentence could turn to *neutral* attitude, so the probability of selecting all '*Neutral*'-related trajectories could be largely increased and probability of selecting all '*Neutral*'-related trajectories could be largely penalised.

In this case, even we increase $W'_B$ by increasing the $N_B$, the substantial enhancement of $W'_B$ still could lead to a larger $V'_B$.

Thus, we can conclude: After pruning branch $C$, the additional simulations allocated to branch $B$ can increase its estimated value due to improved exploration, leading to:

$$V'_B > V'_A$$

$\square$

