# OpenReview forum: "Enhancing LLM Faithfulness in Rationale Generation via Dual-Reward Probabilistic Inference"
_ICLR.cc/2025/Conference — ICLR 2025 Conference Withdrawn Submission_

### Official Review · Reviewer_1TJ5 · 2024-10-21

**Soundness:** 2
**Presentation:** 2
**Contribution:** 2
**Rating:** 5
**Confidence:** 3

**Summary:**

They tackle the rationale generation tasks in LLMs' reasoning process. Specifically, they propose a probabilistic inference paradigm that provides fine-grained and lookahead rewards to instruct LLMs to generate good rationale. The key problem addressed is that LLMs often produce unfaithful explanations, especially when they fail to incorporate essential contextual information.

+ **Local Reward**:  this component ensures coherence with the immediate context, often by using a domain-specific expert model.
+ **Global reward**: This assesses the plausibility of the current token in relation to desirable future attributes

The search algorithm, especially for lookahead reweight seems interesting.

Please forgive me if I misunderstand something. I spent much time for reading the paper but to be honest, I am not an expert in this area. I will available on the rebuttal time for author's response and will read their response. I am also open to other reviewers' opinions.

**Strengths:**

1. The paper introduces a novel probabilistic inference method with a dual-reward mechanism, combining local and global reward. This is a very novel solution.
2. The paper is well-written. I am not an expert in this domain but I can get their core contributions.
3. The experiment design is clear: they design the ablation study in Section 5.1 to justify the local and global rewards for the final performance. Although I suggest authors could do better by choosing more LLMs in different model size to better support their experimental design.

**Weaknesses:**

1. There are several related works that are missing or less discussed:
        + Evaluating Human Alignment and Model Faithfulness of LLM Rationale
        +  On Measuring Faithfulness or Self-consistency of Natural Language Explanations
2. Figure 2 about the distribution of domain-specific words is unclear to me. "showing that our method can respond more actively to those domain-specific words" Why does this part matters to the experimental results.

**Questions:**

1. The overall experiments are conducted on LLaMA3. I think more backbone LLMs and other sizes of LLMs are needed to justify the proposed inference paradigm.
2. More experiments on more related datasets is needed.

---

> ### Author Response · Authors · 2024-11-22
> **Response to Reviewer 1TJ5**
>
> Thank you very much for spending your valuable time in reviewing our work and we address your concerns as follows.
>
> **Q1: Missing references**
>
> **R1**: The paper “Evaluating Human Alignment and Model Faithfulness of LLM Rationale" was released after our paper submission, and we appreciate you bringing it to our attention. This paper focuses on faithfulness explanation from human perspectives and provide important insights for future faithful evaluation. We will add to the our related work discussion.
> We have read the paper “On Measuring Faithfulness or Self-consistency of Natural Language Explanations” when preparing our paper submission. It introduced a method of calculating faithfulness as a continuous value rather than a binary measure. We didn’t cite it as we followed more traditional and widely-used methods for faithful evaluation, as outlined in[1,2,3]. We will add a discussion of this faithfulness metric in the related work section.
>
> **Q2: Explanation of Figure2**
>
> **R2**: For faithfulness, we firstly observed that domain-specific words are important in enhancing context-adherence in Table1, which inspired us to increase the generation probability of domain-specific words to enhance faithfulness of the rationales generated. This motivated the design of our approach using global rewards (expert model). To the best of our knowledge, this is the first time faithfulness has been studied by encouraging domain-specific tokens.
> In the experiment, we show the distribution of domain-specific words for both the backbone model and our method, highlighting that our model successfully generates more domain-specific words (indicated by the blue line above the yellow line).
>
> **Q3: Experiment on other backbone**
>
> **R3**: We have updated the evaluation results on Mistral 7B, shown in Appendix B1. The evaluation results shows that our method can also achieves better results in accuracy and better faithfulness in rationale.
>
> **Q4: Experiment on other datasets**
>
> **R4**: Thanks for your suggestions. We have tested our method across three distinct tasks, i.e., student essay assessment, natural language inference and question answering, using 7 different datasets. The consistent improvements across these tasks  validate the effectiveness of our methods, with an absolute accuracy improvement of 33% across the seven datasets, along with a 10% improvement in faithfulness evaluation.  Please kindly let us know if you have any particular datasets you recommend for further evaluation.
>
> Please let us know if you have further concrete questions or concerns that we can address. Thank you for your engagement with our work.

---

> ### Author Response · Authors · 2024-11-25
>
> The discussion that we can participate in will end soon. Could you kindly confirm whether our responses and the revision in the newly uploaded pdf have appropriately addressed your concerns? If you find that we have properly addressed your concerns, we kindly request that you consider adjusting your initial score accordingly. Please let us know if you have further comments.
>
> Thank you for your valuable time and effort in reviewing our work.

---

> > ### Comment · Reviewer_1TJ5 · 2024-11-26
> >
> > Thanks for your response. I read it while I'd like to maintain my score.
> >
> > Thanks

---

### Official Review · Reviewer_MSQh · 2024-11-01

**Soundness:** 2
**Presentation:** 1
**Contribution:** 1
**Rating:** 3
**Confidence:** 3

**Summary:**

This work proposes an inference-time method to improve the performance and faithfulness of general (instruction-tuned) large language models (LLMs). Specifically, the method uses expert models to provide fine-grained and lookahead rewards to search and reweight possible tokens or continuations proposed by the LLM. With the help of expert models trained on the target task or domain, the proposed method can improve both the accuracy and faithfulness of the zero-shot answers of two instruction-tuned models on three reasoning tasks.

**Strengths:**

The direction this paper explored has been receiving increasing interest recently: improving the quality of LLM answers at inference time without modifying the model weights directly. The proposed method improves the zero-shot accuracy and faithfulness of two strong general instruction-tuned models (Llama-3-8B and Mistral-7b-Instruct-v0.3) on three reasoning tasks. The experiment showing the benefits of going beyond local/token-level rewards and taking into account the global/lookahead reward is interesting.

**Weaknesses:**

- There needs to be more details explaining the proposed method, the motivation of each part, the equations and variables, the relation to related work, and the implementation details. Specifically:
  - Section 3.3: how does the Feynman-Kac Formulae model inspire the faithfulness-seaking search framework? The connection is not straightforward. The notation of eq 1 is ambiguous. What does posterior P_t(st) mean exactly? How is it used in the proposed method? Also, the equation itself needs more explanations on what it is computing and why in this way.
  - Section 3.4 (Local constraint): line 179 I find it hard to follow the motivation. How "certain attributes can be implicitly conveyed over longer spans rather than the individual token" is connected to "Instead, domain-specific experts tend to demonstrate better accuracy in knowledge-rich tasks."? If the domain expert has better accuracy why not just use the expert to predict the scores? Why bother to use them to improve the backbone LLM? In lines 180-181, it says "we introduce a set of classification label words C from these expert models ...", how is C constructed? What is the motivation behind token masking?
  - Section 3.4 (Lookahead Reweight): Equation 3 is hard to understand without proper explanations. $m$ and $x_i$ are not explained in the texts. $s_{t+l}=s_{t-1}||w_t$ is more confusing: $s_{t+l}$ has $t+l$ tokens while $s_{t-1}||w_t$ has $t$ tokens. What does equality mean here?
- Many experimental details are missing, and important experiments are missing.
  - Missing baselines: the performance and faithfulness of the expert models alone. If the faithfulness or accuracy of the expert models are better than the backbone LLM, why do we even need to use the expert models to improve the backbone LLM?
  - Evaluation details: how is the original model evaluated? If it is a zero-shot evaluation. What is the exact prompt and task format used? How to extract answers from the outputs to calculate the accuracy? The backbone LLMs are state-of-the-art instruction-tuned models. However, the task performance as well as the faithfulness are quite low, so the authors need to provide more details on the evaluation.
  - What is the choice of hyperparameter n (number of rollouts) and how is it chosen?
- The writing of the paper could be improved for better readability. First, the paper is not properly scoped. For example, in lines 16-18, it says "... to ensure that LLM-generated rationales are logically coherent and comprehensive." However, there is no result discussing the logical coherence or comprehensiveness of answers in the paper. Another example is line 108: it says "We firstly introduce the faithfulness definition in our context,", but there is no clear definition in section 3.2.

**Questions:**

1. If the expert model is as big as the base model, how can the computational cost be similar to beam search?

---

> ### Author Response · Authors · 2024-11-22
> **Response to reviewer MSQh (1)**
>
> Thank you for your valuable time and your detailed feedback! We address each point as follows.
>
> **Explanation of the proposed method**
>
> The equations referenced are primarily derived from the probabilistic framework of the Feynman-Kac model. We have significantly revised Sections 3.3 and 3.4 in the manuscript, along with an updated Algorithm 1 that introduces our method in a step-by-step manner. Please refer to the revised PDF for detailed explanations. Below is a pointwise response to your proposed questions:
>
> - **Q1: Motivation of applying Feynman-Kac Formulae in the faithfulness-seaking search framework?**
>
> - **R1**: We refer to the Feynman-Kac framework, which is a general framework of incorporating a potential function to adjust the original conditional probability. For faithfulness, we observed that the expert model tends to generate more domain-specific tokens, which contribute to context-coherent and faithful rationales. This insight motivated us to employ the expert model’s conditional generation probability as the potential function to adjust the backbone model’s conditional generation probability accordingly.
>
> - **Q2: Explanation of Eq1**.
> - **R2**: $\mathbb{P}_t​(s_t​)$ represents the probability of reaching $s_t$ from the $\mathbb{P}_t$. Specially, $[S_t=s_t]$ is an indicator function that is equal to 1 if the state at $t$ is $s_t$, and 0 otherwise. The numerator inside the expectation represents the product of rewards and the probability of reaching state $s_t$, ensuring that paths leading to high rewards over time are given more weight. For better understanding, we connect with concepts in MCTS. **Rollout in MCTS** is used to estimate the value of future actions, helping navigate and expand the search tree effectively. The Eq.(1) represents a probabilistic reward distribution of state $s_t$. $G_t$ is analogous to reward function in MCTS, with superiority in estimate the reward via lookahead. In our framework, the global expert model $U^g$ performs reward estimation to the generated policy from backbone model (see `GlobalReward` function).
>
> - **Q3: Explanation of line 179 about motivation of local constraint.**
> - **R3**: *Firstly*, we would like to emphasise that attributes in language, such as toxicity and faithfulness, can manifest across longer spans of text rather than being confined to explicit attribute-bearing words. This is why we opt not to utilize token-level constraints like logit fusion [1] for addressing the faithful rationale generation problem. Such methods may fall short in capturing the nuanced and distributed nature of these attributes over extended contexts. *Instead*, we align with existing literature highlighting the effectiveness of domain-specific experts in achieving higher accuracy on tasks rich in domain knowledge. For example, task-specific classifiers are known to excel in tasks requiring specialised expertise. This observation inspires our approach of incorporating domain experts for label prediction to enhance both accuracy and faithfulness in rationale generation.
>
> - **Q4: why not just use the expert to predict the scores.**
> - **R4**: Thanks for raising this interesting point! (a) The primary goal of our framework is to maintain a trade-off between task performance (accuracy) and rationale faithfulness. Incorporating predicted results directly as prompts can sometimes lead to issues like hallucination and irrelevant context generation, as observed in models like ChatGPT [2]. (b) From the results in Table 5 and Table 6, it is evident that relying solely on local or global expert guidance fails to strike the required balance. This underscores the importance of our dual-reward framework, which combines both local (answer accuracy) and global (faithful rationale) rewards. Our approach ensures the LLM generates outputs that are not only accurate but also aligned with domain-specific knowledge. (c)  Our framework integrates expert predictions in a probabilistic manner, i.e., treating the prediction as a conditional input within the generation process. This will fundamentally guide the rationale generation process and is likely to produce a rationale that is more consistent and faithful to the prediction.
>
> - **Q5: local contraint**
> - **R5**: For better clarity, we remove the Eq2 and Eq3. The calculation of Local reward is as follows: We introduce a set of classification label words $\mathcal{C}$ and we remove the label words which are not included in the expert model’s prediction $c_0$. We then renormalise the output probability based on the new vocabulary (see function `LocalMask` for details).

---

> > ### Comment · Reviewer_MSQh · 2024-11-26
> >
> > I appreciate the authors' effort in addressing my concerns and revising the paper accordingly. Still, most of my concerns remain.
> >
> > Algorithm 1:
> >
> > 1. inconsistant notation, what is $f$ and $f_\theta$? Why in local mask sometime $\pi$ sometime $\pi_t$?
> >
> > > We refer to the Feynman-Kac framework, which is a general framework of incorporating a potential function to adjust the original conditional probability. For faithfulness, we observed that the expert model tends to generate more domain-specific tokens, which contribute to context-coherent and faithful rationales. This insight motivated us to employ the expert model’s conditional generation probability as the potential function to adjust the backbone model’s conditional generation probability accordingly.
> >
> > I don't see how is this reflected in Algorithm 1. In the while loop all tokens are directly sampled from $\pi_t$ without any adjustment, and looks like the algorithm is just sample K examples and choose the sample with the highest probability according to the expert model's distribution. I don't see how Algorithm 1 is connected to Feynman-Kac model and why is it a search algorithm.
> >
> > > *Firstly*, we would like to emphasise that attributes in language, such as toxicity and faithfulness, can manifest across longer spans of text rather than being confined to explicit attribute-bearing words. This is why we opt not to utilize token-level constraints like logit fusion [1] for addressing the faithful rationale generation problem. Such methods may fall short in capturing the nuanced and distributed nature of these attributes over extended contexts. *Instead*, we align with existing literature highlighting the effectiveness of domain-specific experts in achieving higher accuracy on tasks rich in domain knowledge. For example, task-specific classifiers are known to excel in tasks requiring specialised expertise. This observation inspires our approach of incorporating domain experts for label prediction to enhance both accuracy and faithfulness in rationale generation.
> >
> > I still don't understand why Localmasking would benefit faithfulness. To me it look like a way to improve the accuracy. **So seems like the whole algorithm is to use domain-expert to predict the correct label, then use the large backbone model to generate some candidate explanations, and use another faithful expert to score them.** Also my concern of how the class label words $C$ are chosn still remains.
> >
> > > The primary goal of our framework is to maintain a trade-off between task performance (accuracy) and rationale faithfulness. Incorporating predicted results directly as prompts can sometimes lead to issues like hallucination and irrelevant context generation, as observed in models like ChatGPT [2].
> >
> > Let the backbone model answer directly can also potentially hallucinate. I don't understand why localmasking will hallucinate less than directly use domain-expert to predict the label. As for irrelevant context generation one can simply do as in local masking to restrict the domain-expert's prediction only on the label words for extracting the answer.
> >
> > > (b) From the results in Table 5 and Table 6, it is evident that relying solely on local or global expert guidance fails to strike the required balance. This underscores the importance of our dual-reward framework, which combines both local (answer accuracy) and global (faithful rationale) rewards. Our approach ensures the LLM generates outputs that are not only accurate but also aligned with domain-specific knowledge.
> >
> > 1. The definition of the term "CLS" is removed from the revised paper, where it used to be the first sentence of section 5.1.
> > 2. I don't understand your point. From Table 5 apparently CLS (Expert model) perform better than your method on all datasets? It is also mentioned in the paper "Notably, when incorporating the CLS, our method does not necessarily perform as well as the classifier alone." So why not just use CLS to predict the label word? Anyway your method uses domain experts, so why not choose the most accurate model for generating the answer? The faithfulness of the rationale can be addressed separately.

---

> ### Author Response · Authors · 2024-11-25
> **Response to reviewer MSQh (2)**
>
> **Missing comparison with expert model**
>
> Thank you for pointing out this critical point. The comparison results are updated in Table 5 and Table 6, where **CLS** is the expert model used to predict the answer and the **expert model** refers to the global expert model used to generate rationale. As CLS is a classifier that can't generate rationale, we only apply it in acc evaluation, while the expert model is applied in both metrics calculation. Below is our analysis:
>
> **(a) comparing with fine-tuned backbone model**:
>    We have updated the ablation results in Table 5 and Table 6. For the ASAP dataset, the expert model uses the same backbone as the primary model (i.e., Llama3-8B) and has been fine-tuned on the ASAP training set. This aligns well with your intended comparison with *Vanilla Fine-Tuning of Backbone Model*
>    - **Accuracy Results**: In Table 5, by comparing the expert model with our full framework, we observe overwhelming advantages of our full model across **all four subsets**, especially on Q2, with results showing 68% vs. 48% (ours vs. expert). And the average results are 74\% and 69\% for our (full) and expert model.
>    - **Faithfulness Results**: The expert-only method achieves better faithfulness on three subsets (except for Q4). Interestingly, our full framework (including CLS & expert) behaves better than our+expert model, showing the synergised effects of CLS and expert.
> - **Overall Results**: Although our method fails to exhibit better faithfulness compared with the fine-tuned model expert model, it instead strikes a challenging trade-off between accuracy (CLS) and faithfulness (expert model), which has been discussed in [1]. This is also one of the core motivations: combining the strengths of classification-specialised and rationale-specialised models.
>
> **(b) compare with smaller fine-tuned models**:
> For other datasets where the expert models are Llama2-7B (smaller than our backbone): (i) The comparison between our full (i.e., *our+cls+expert*) and *our+expert* also shows that the synergistic effects of combining the two experts. (ii) the faithfulness for our (full) is better than the expert model on SNLI and MNLI, with 15\% vs 13\% and 19\% vs 9\%, respectively. This suggests that our approach not only achieves faithfulness improvements over smaller expert models but also has the potential to leverage weak supervision to unlock the capabilities of larger backbone models.
>
> **(c) Generalised to Out-of-Task Expert Model**:
> Note that it is not strictly necessary for the expert model to be trained on the exact task dataset. For instance, we experimented with Expert Model 2 (/Weyaxi/Einstein-v2-7B in huggingface), which is trained on general science question-answering instead of ASAP-specific data. Results in Table 12 show that incorporating this out-of-task expert model leads to improved faithfulness on 11 out of 12 metrics. These results validate the generalisability of our method.
>
> Overall, our framework demonstrates (i) superior performance in balancing accuracy and faithfulness when the expert model is of the same size, (ii) improved faithfulness compared to smaller, in-task trained expert models, and (iii) robust generalizability, effectively adapting to scenarios where the expert model has not been specifically trained for the inference task.
>
> **Evaluation details**:
> Thanks for asking the question about detailed evaluation, which makes our method clarified. We have updated the experiment setup section in Appendix A, with prompt used in different tasks, and the hyperparameter configuration.
>
> **Q1: The exact prompt and task format used**
> Yes, we use zero-shot evaluation. Please refer to the prompt template in Appendix A.
>
> **Q2: How to extract answers from the outputs to calculate the accuracy?**
> The answer/predicted labels are extracted using specifically designed regular expressions corresponding to our format requirements within our prompt instruction.
>
> **Q3: Explanation of the undesirable performance of backbone models**
> The backbone model performs lower since those models are not directly trained on our selected datasets. This is also evident in the LLaMA paper [2] as well, which shows that zero-shot inference on TruthfulQA with the LLaMA 7B model has an accuracy of 33\%.
>
> **Q4: number of rollouts**
> We use a beam of 3 and 10 particles for decoding. Therefore, 30 different rollouts are calculated during decoding.
>
> **References**:
> [1] Question Decomposition Improves the Faithfulness of Model-Generated Reasoning. ICML23
> [2] LLaMA: Open and Efficient Foundation Language Models. 2023

---

> > ### Comment · Reviewer_MSQh · 2024-11-26
> >
> > > Overall Results: Although our method fails to exhibit better faithfulness compared with the fine-tuned model expert model, it instead strikes a challenging trade-off between accuracy (CLS) and faithfulness (expert model), which has been discussed in [1]. This is also one of the core motivations: combining the strengths of classification-specialised and rationale-specialised models.
> >
> > Thanks for adding the expert model results. In this case, I don't see a clear benefit of the proposed method over the expert models. As for the argument that the proposed method achieves a trade-off between accuracy and faithfulness, one can easily combine CLS and Expert by first predicting label words using CLS then explaining the answer using the global expert model. I feel this should be a strong baseline and would at least perform as well as the proposed method in achieving the balance between accuracy and faithfulness.

---

> ### Author Response · Authors · 2024-11-25
>
> We've taken your initial feedback into careful consideration and incorporated them into our revised Pdf as indicated in the **Summary of Revision**. Could you kindly confirm whether our responses have appropriately addressed your concerns? If you find that we have properly addressed your concerns, we kindly request that you consider adjusting your initial score accordingly. Please let us know if you have further comments.
>
> Thank you for your time and effort in reviewing our work.

---

> ### Author Response · Authors · 2024-11-28
> **Response to Reviewer MSQh**
>
> We sincerely appreciate the reviewers' valuable time and thoughtful feedback, which have provided us with an opportunity to further clarify our contributions and strengthen our work.
>
> > Inconsistant notation, $f_\theta$, $\pi_t$
>
> We totally agree that consistency and clarity are very important, so we have updated the PDF to keep the complete form. Actually, We use $f_\theta$ refers to a parameterised function, explicitly highlighting that $\theta$ are the parameters. While in Algorithm 1, the parameters are not the focus, so we use $f$ throughout the algorithm. For $\pi_t$ and $\pi$ in the localmask function, we omit the $t$ as we have the input within the function $\pi$, such as $\pi(x_t=v’|x_{t-1})$.
>
> > How Algorithm connected to Feynman-Kac and search algorithm
>
> **Q1:  No Adjustment to $\pi$?**
>
> **R1** The Feynman-Kac framework is a classical framework proposed in 2024 and serves as a general foundation for probabilistic inference. Its key idea is to introduce the potential function $G_t(s_{t-1}, s_t, f_\theta)$, which acts as a reward function to score the current state. Some methods modify the policy $\pi_t$​ to derive a new policy $\pi’_t$. An example is our localmask method. In contrast, our global reward mechanism reweights the output probabilities using an expert model, allowing tokens with relatively low probabilities to be sampled while eliminating high-probability tokens. In that way, we could change the selected topk tokens according to the score from expert model. **Based on the changes in the output sequence at step $t$, the backbone model will generate differently at timestep $t+1$, as the conditional generation depends on the updated conditioned input.** It’s worth noting that there are numerous practical approaches to modifying output distributions. The core idea of the Feynman-Kac framework is to employ a function with lookahead characteristics, which helps avoid local optima.
>
> **Q2: Why search algorithm?**
>
> **R2**: The existing Monte Carlo Tree Search (MCTS) algorithm, utilizes a policy model to generate multiple candidate nodes. These nodes are then evaluated by a reward model, and the state (node) with the highest reward is selected for tree expansion. Similarly, in our Algorithm 1, at each timestep, the backbone model generates K tokens (where K is the beam search size). These tokens are then evaluated by the expert model. Among the $K \times N$ generated trajectories, only the trajectory with the highest reward is selected. While our search process does not exactly replicate the step-by-step node selection and expansion in MCTS, it generates multiple trajectories in parallel to reduce computational cost and it should still be categorised as a MCTS algorithm, such as the paper [1][2].
>
> > I still don't understand why Localmasking would benefit faithfulness. To me it look like a way to improve the accuracy.
>
> Yes, your understanding is totally correct. Local masking (CLS) benefits the accuracy, while global reward (expert) benefits the faithfulness.
>
> > So seems like the whole algorithm is to use domain-expert to predict the correct label, then use the large backbone model to generate some candidate explanations, and use another faithful expert to score them.
>
> In general, yes, we use a domain-expert model, a classifier (CLS), to ensure prediction accuracy, and a generative expert model to enhance faithfulness. However, we would like to clarify two key points:
>
> - The predicted label from the CLS is not directly used as the final label.
> - The expert model is applied step-wise rather than after the sequence generation is complete.
> These differences can lead to distinct outcomes. For example, we generate 3 tokens at each timestep $t-1$, the step-wise approach ensures that the top-K tokens are selected based on the expert model’s score, denoted as $w_t^{e}$​ (e.g., *apple*, *orange*, *peach*). This selection may differ from the top-K tokens determined solely by the backbone model’s output distribution, $w_{t-1}^{b}$​ (e.g., *this*, *the*, *that*). Each token in $w^{e}_{t-1}$ is then incorporated into the conditional context to guide the backbone model’s generation at $t$. For instance, the generation at $t$ would be conditioned on that is $\pi_t(x_t|\text{apple})$, rather than is $\pi_t(x_t|\text{this})$. Thus, the expert model provides process-level, fine-grained controllability over the generation, rather than a post-hoc adjustment.
>
> > how the class label words $\mathcal{C}$ are chosen still remains.
>
> Our label words $\mathcal{C}$ are selected based on the datasets’ label space, see the details in Line 238-239. Please let us know if you have further concrete question.
>
> > CLS The definition of the term "CLS" is removed from the revised paper, where it used to be the first sentence of section 5.1.
>
> Thank you for noticing the change! We apologise for this mistake and have corrected accordingly in the newly updated revised version.

---

> > ### Comment · Reviewer_MSQh · 2024-12-02
> >
> > Thank you for the explanations. I am still not quite convinced that the current section 3 and algorithm 1 describe the proposed method clearly and in detail enough:
> >
> > >  In contrast, our global reward mechanism reweights the output probabilities using an expert model, allowing tokens with relatively low probabilities to be sampled while eliminating high-probability tokens. In that way, we could change the selected topk tokens according to the score from expert model. Based on the changes in the output sequence at step t, the backbone model will generate differently at timestep t+1, as the conditional generation depends on the updated conditioned input.
> >
> > I am still not quite sure how this is reflected in Algorithm 1.
> > 1. The global reward $\alpha_t^k$ are only used for selecting the output sequence after all the sampling is done, and the global reward does not affect the sampling stage at all. To me this is just using a reward model to weight/score K individually sampled sequence.
> > 2. "Based on the changes in the output sequence at step t, the backbone model will generate differently at timestep t+1, as the conditional generation depends on the updated conditioned input." According to Algorithm 1: $x_{t+1}^k\sim \pi_t(x_{t+1}^k|x_{t+1}^k, f_\theta)$ for $k=1,\dots,K$, the K sequences are sampled individually, so this is just parallel sampling K length T sequences. I don't know how is adjustment-of-probability aspect of Feynman-Kac reflected here. Also, I don't think it is necessary to motivate using an expert model to score parallelly sampled outputs with something as complicated as Feynman-Kac.
> >
> > > Similarly, in our Algorithm 1, at each timestep, the backbone model generates K tokens (where K is the beam search size). These tokens are then evaluated by the expert model. Among the $K\times N$ generated trajectories, only the trajectory with the highest reward is selected. While our search process does not exactly replicate the step-by-step node selection and expansion in MCTS, it generates multiple trajectories in parallel to reduce computational cost and it should still be categorised as a MCTS algorithm, such as the paper [1][2].
> > 1. "at each timestep, the backbone model generates K tokens (where K is the beam search size)": each of the $K$ sequence are generated independently, so I am not sure why is K called beam search size.
> > 2. "Among the $K\times N$ generated trajectories", if I understand correctly, there are only $K$ length-$T$ trajectories.
> > 3. I am not convinced that it is proper to call a method that just scores parallelly sampled outputs as a search algorithm if the reward model does not affect how the trajectories are sampled.
> >
> > Nevertheless, I appreciate your effort in addressing my concern and improving the paper.

---

> ### Author Response · Authors · 2024-11-28
>
> > Let the backbone model answer directly can also potentially hallucinate. I don't understand why localmasking will hallucinate less than directly use domain-expert to predict the label. As for irrelevant context generation one can simply do as in local masking to restrict the domain-expert's prediction only on the label words for extracting the answer.
>
> **R1**: According to the response above. Our method is not equal to let the backbone answer directly, the expert model does affect step-wise generation.
>
> **R2**:  Localmask is essentially classifier (not for generation task, so that they can't generate label words), and they are more likely to generate highly accurate prediction than generative models, i..e, expert model, as shown in Table 5. Therefore, we do not use the generative model for label prediction.
>
> > I don't understand your point. From Table 5 apparently CLS (Expert model) perform better than your method on all datasets? It is also mentioned in the paper "Notably, when incorporating the CLS, our method does not necessarily perform as well as the classifier alone." So why not just use CLS to predict the label word? Anyway your method uses domain experts, so why not choose the most accurate model for generating the answer? The faithfulness of the rationale can be addressed separately.
>
> Sorry for the confusion.
> - First, we would like to clarify that our framework utilizes two types of expert models for local and global rewards, respectively. The local reward model, referred to as CLS in our experiments, is a classifier (not a generative model) and serves as the primary source of accuracy by scoring prediction consistency. The global reward model, referred to as the expert model, is a generative model and primarily contributes to enhancing faithfulness in the generated outputs.
> - Second, if I understand correctly, the idea is to use two models—one aimed at accurate answer generation and the other at maintaining faithfulness. We acknowledge that improving both accuracy and faithfulness within a single model remains a key challenge, as emphasized in recent works (e.g., [1]). Moreover, even if we had access to accurate labels and directly fed them into the generative model, this approach would not necessarily guarantee improvements. On the contrary, incorporating predicted results directly as prompts can introduce issues such as ungrounded hallucinations, as highlighted in recent studies [2, 3]. Thus, effectively balancing accuracy and faithfulness continues to be a non-trivial and active area of research.
>
> Please let us know if you have any concerns about our motivation behind or any other concrete questions.
>
> Thanks again for your patience and valuable time in enaging in our work!
>
> **References**
>
> [1] Question Decomposition Improves the Faithfulness of Model-Generated Reasoning.
>
> [2] Siren's Song in the AI Ocean: A Survey on Hallucination in Large Language Models.
>
> [3] Chain of Natural Language Inference for Reducing Large Language Model Ungrounded Hallucinations

---

> ### Author Response · Authors · 2024-11-28
>
> > Thanks for adding the expert model results. In this case, I don't see a clear benefit of the proposed method over the expert models. As for the argument that the proposed method achieves a trade-off between accuracy and faithfulness, one can easily combine CLS and Expert by first predicting label words using CLS then explaining the answer using the global expert model. I feel this should be a strong baseline and would at least perform as well as the proposed method in achieving the balance between accuracy and faithfulness.
>
> Thanks for raising this critical point! Your question about the performance about backbone model inspired us to rethink the evaluation results. As the model are differently sensitive to the text, so the faithfulness metrics vary lot across different datasets, for example, from 0.01 to 0.05 on ASAP, and around 0.1 for NLI task. Consequently, we have provided a normalised faithfulness evaluation table below. The faithfulness score is scaled by the maximum and minimum scores in each task.
>
> **Table1: Normalised faithfulness evaluation**
> |            | Backbone | Expert | Our(Full) |
> |:----------:|:--------:|:------:|:---------:|
> | ASAP-1     | 0.2920   | 0.8230 | 0.4513    |
> | ASAP-2     | 0.4425   | 1      | 0.4336    |
> | ASAP-3     | 0.3628   | 0.5310 | 0.5044    |
> | ASAP-4     | 0        | 0.4602 | 0.8938    |
> | SNLI       | 0.2000   | 0.4000 | 0.6000    |
> | MNLI       | 0        | 0      | 1         |
> | TruthfulQA | 0        | 1      | 0.7273    |
> | Average    | 0.1853   | 0.6020 | 0.6586    |
>
> Our full method achieves the highest faithfulness score compared to using the backbone or the expert model alone.
>
> Then, we provide an average faithfulness-accuracy comparison in Table2.
>
> **Table2: Averaged normalised faithfulness-accuracy**
>
> |           | Normalised Faithfulness | Accuracy |   Sum  |
> |:---------:|:-----------------------:|:--------:|:------:|
> |  Backbone |          0.1853         |  0.4171  | 0.6025 |
> |   Expert  |          0.6020         |  0.6914  | 1.2935 |
> | Our(full) |          0.6586         |  0.7400  | 1.3986 |
>
> Our(full) achieved the highest in a combination of accuracy and faithfulness evaluation, with clear improvements.
>
> Furthermore, we are in the process of implementing the baseline you mentioned—using the label predicted by CLS as a prompt for the rationale generation model. We will update our results accordingly once this implementation is completed.
>
> Thank you again for your valuable suggestions. We are truly grateful for your feedback, which has significantly improved the clarity of our paper. We sincerely hope that the adjustments and clarifications we have provided address at least part of your concerns. If so, we kindly request that you consider adjusting your scores to reflect our efforts in responding to your feedback as a positive signal. Also, please let us know if you have further more suggestions!

---

> > ### Comment · Reviewer_MSQh · 2024-12-02
> >
> > Thank you for providing additional results and discussions. Still
> > 1. I am not sure if **averaging** the normalized faithfulness is the correct way to compare the proposed method with the baselines. In fact, the expert model is more faithful in 4 out of 7 datasets, and the most contribution to the advantage of the proposed method seems to come from MNLI.
> > 2. As for the accuracy, I am not sure why we are comparing the expert model rather than the CLS baseline.
> > 3. I still feel CLS + expert model would be a very strong baseline in the setting of this paper.
> >
> > Overall I feel the experiments need to be conducted and analyzed more carefully, and the proposed algorithm needs to be better motivated, scoped, and described in detail. As a result, I think the current paper is not yet ready for publishing, and I would maintain my current evaluation score.

---

> ### Author Response · Authors · 2024-12-03
>
> Thank you once again for your detailed feedback and thoughtful comments!
>
> We would like to emphasise that our expert model can select different $x_{t-1}$, based on the $x_{t-1}$, the backbone model which is responsible for trajectory generation will generate/sample differently as the generation probability is $p(|x_{t-1})$. Details in the response to `why search algorithm` above.
>
> We greatly appreciate your valuable feedback, which has been instrumental in refining our work. We will continue improve our paper!
>
> Thank you!

---

### Official Review · Reviewer_msmv · 2024-11-04

**Soundness:** 2
**Presentation:** 1
**Contribution:** 2
**Rating:** 1
**Confidence:** 3

**Summary:**

The work aims to improve the faithfulness of the LLM-generated rationales for reasoning tasks. They propose an inference-based method where an LLM is guided to generate more faithful rationales by both local and global rewards. Both rewards are provided by additional expert models which are trained on the downstream tasks. Experiments demonstrate the effectiveness of the method in achieving higher accuracy and faithfulness.

**Strengths:**

1. Faithful rationales are important for explainability and model control, which makes this work well-motivated.
2. The proposed method is training-free (although with reliance on trained expert models), making their method portable.
3. A comprehensive set of experiments is conducted to showcase the effectiveness of their proposed method.

**Weaknesses:**

1. The method requires the model to generate the answer prior to the rationale, which provides no guarantee that the decision is made based on the rationale. The model could still suffer from inherent biases.
2. The method is limited to reasoning tasks with constrained answer space, limiting its generalization to more open-ended tasks.
3. The method is poorly introduced. It would be very helpful if the authors could explain what exactly Eq.1-3 are doing in plain words.

**Questions:**

Could this method generalize to the setting where the rationale is generated before the answer?

---

> ### Author Response · Authors · 2024-11-22
> **Response to Reviewer msmv (1)**
>
> Thanks for reviewing our work and we address your concern in the generalisability of our framework.
>
> **Q1: Generalise to situations where the rationale is generated before the answer.**
>
> *R1:*  It is certain that LLMs can generate answers either before or after the rationale.
> - (i) To prevent scenarios where an overly long rationale causes the answer to exceed the output length limit, we prioritize generating the answer first. This is achieved by providing explicit instructions and demonstrations where the answer precedes the rationale, ensuring the backbone LLM generates the answer at the beginning.
> - (ii) In our current setting, this approach is motivated by the observation that specialised smaller models often perform better at classification tasks. By leveraging these models for accurate rating, we establish a prior of a likely correct rating, which in turn helps ensure that the rationale generated afterward is more faithful to the truth.
> - (iii) Through **Algorithm 1** updated in Section 3.3, our incorporation of the function `localmask` for answer prediction is implemented at the first timestep. Meanwhile, the rationale reward is applied throughout the sequence. This design allows for the `localmask` to also be applied at the final timestep, guided either by a length limit or a terminal token indicator. Additionally, we need to evaluate whether feeding the generated rationale back into the classifier via the `localmask` function would degrade or enhance classification accuracy.
>
> **Q2:  Limited tasks to reasoning.**
>
> **R2**: Thank you for raising this critical point.
>
> Firstly, we would like to clarify that existing faithfulness evaluations are traditionally based on the assumption that answers can be straightforwardly evaluated as either identical or not. Faithfulness evaluation primarily focuses on determining whether changes in the input lead to corresponding changes in the answer. Evaluating open-ended questions introduces the additional challenge of assessing semantic equivalence, which is not the primary focus of most existing studies on faithful rationales. For example, research in this area often evaluates tasks with clear answer spaces, such as Natural Language Inference (NLI) and multiple-choice QA [1] (both of which are included in our evaluation), as well as sentiment classification [2] and fact-checking tasks framed as binary classification [3].
>
> Secondly, our method is extendable to scenarios with an infinite label space $( |\mathcal{C}| = \infty )$, even though the current evaluations are conducted on tasks with a constrained label space $( |\mathcal{C}| = N \in \mathbb{N} )$. For instance, in mathematical problem-solving tasks, the answer could be any arbitrary number. In such cases, the expert model provides a prediction $M$, with its confidence expressed as the probabilities $ w_1$ for $M $ and  $w_2 $ (for the second most probable prediction). The ratio $\frac{w_1}{w_2} $ serves as an indicator of the expert's confidence in $M$ [4]. This confidence is then used as a multiplier to enhance the backbone model's prediction for $M$. Finally, the backbone model's transition distribution is renormalised to ensure a valid probability distribution.

---

> ### Author Response · Authors · 2024-11-22
> **Response to Reviewer msmv (2)**
>
> **Q3: Eq.1-3 are doing in plain words**.
>
> **R3**:  The equations referenced are primarily derived from the probabilistic framework of the Feynman-Kac model. We have significantly **revised Sections 3.3 and 3.4 in the manuscript**, along with an updated **Algorithm 1** that introduces our method in a step-by-step manner. Please refer to the revised PDF for detailed explanations.
>
> Additionally, we provided a general response to all reviewers, detailing how the two rewards—local and global—are calculated (Eq. 2 and Eq. 3). Below is a pointwise response to your proposed questions:
> - Eq1: In the context of generating tokens $s_t$ using model $f_\theta$, The potential function $G_t$ maps $(s_t, s_{t+1})$ to a non-negative score, analogous to the reward function. The adjusted probability of $f_\theta$ generates $s_t$ is calculated Eq1. $[S_t=s_t]$ is an indicator function that is equal to 1 if the state at $t$ is $s_t$, and 0 otherwise. The numerator inside the expectation represents the product of rewards and the probability of reaching state $s_t$, ensuring that paths leading to high rewards over time are given more weight.} Generation continues until a terminal token or the maximum length of the sequence $T$, i.e., $t \wedge T=\text{min}(t,T)$
> - Local Reward (Eq2): We introduce a set of classification label words and we remove the label words which are not included in the expert model’s prediction. We then renormalise the output probability based on the new vocabulary.
> - Global Reward (Eq3):  The expert model generates a lookahead reward by evaluating the plausibility of the tuple $(x_t, x_{t+1})$ generated from the backbone model.
>
>
> **References**
>
> [1] Faithfulness tests for natural language explanations. ACL2023
>
> [2] Faithful explanations of black-box nlp models using llm-generated counterfactuals. ICLR24
>
> [3] Can llms produce faithful explanations for fact-checking? towards faithful explainable fact-checking via multi-agent debate. 2024
>
> [4] Confidence-aware learning for deep neural networks. ICML2020
>
>
> Please let us know if you have further questions or any feedback. Thanks!

---

> ### Author Response · Authors · 2024-11-25
>
> We've taken your initial feedback about (i) "the generalisability of method" (ii) writing about method into careful consideration and incorporated them into the revised pdf. Could you kindly confirm whether our responses have appropriately addressed your concerns? If you find that we have properly addressed your concerns, we kindly request that you consider adjusting your initial score accordingly. Please let us know if you have further comments.
>
> Thank you for your time and effort in reviewing our work.

---

> > ### Comment · Reviewer_TvLG · 2024-11-26
> > **Thanks for the response**
> >
> > I have bumped up the score since many of my questions were addressed satisfactory. I am still not very convinced by your discussion on the novelty aspect. I'm not sure if the highlighted difference with ILQL is very significant for example.

---

> ### Author Response · Authors · 2024-11-27
> **Response to Reviewer TvLG: Follow-up Questions**
>
> Many thanks for recognising the improvements in our updated version and for your valuable feedback! We also appreciate your constructive suggestion to further clarify the distinction between our method and ILQL.
>
> Firstly, at a high level, both methods share the similarity of leveraging logit perturbation to enhance the output of the backbone model. For clarity, we denote the backbone model as $M_b$ and the expert model as $M_p$.
>
> Then, we provide a detailed comparison in the following aspects:
>
> **Model Types**
> - ILQL: Both the backbone model (referred to as the standard language model) and the perturbed model are GPT-2 small.
> - Ours: Our backbone models are LLaMA3-8B and Mistral-7B, while perturbed models include LLaMA3-8B, LLaMA2-7B, and Mixtral-7B. Importantly, our method supports backbones and perturbed models of differing sizes and tokenisations.
>
> **Does it require extra training before deployment?**
> - ILQL: Yes, there are three types of $M_{p}$: policy generation $\pi_B$, value function model for $Q$ and $V$ generation, and a target value network. They are trained via implicit Q-learning objectives on the exact dataset same as inference tasks.
> - Ours:  No, we can use any $M_{p}$ pretrained on in-task datasets, or even out-of-task datasets (see response above `Fair Comparison with Vanilla Fine-Tuning of Backbone Model`.)
>
> **Perturbation strategy**:
> - ILQL:  calibrate the original policy using trained $Q$ and $V$, i.e., $\pi(a \mid h) \propto \pi_\beta(a \mid h) e^{\beta(Q(h, a) - V(h))} = \exp(\log(\pi_\beta(a \mid h)) + \beta(Q(h, a) - V(h))).$
> - Ours: the step-wise reward is given by $M_p$, and then select the sequence with highest sentence-level reward from generated $K$ sequences (see in Algorithm 1). This reward strategy is inspired by our preliminary study in Figure 1 that $M_p$ prefers in-domain text and could contribute to faithfulness.
>
> **Overall analysis**
> - Our method can fit the situation where $M_b$ and $M_p$ have different but larger backbones. In contrast, ILQL's reliance on training three models for $M_p$ can limit its scalability to larger backbone models, such as fine-tuning a LLaMA3-8B. Our approach, however, can leverage any expert model, even if pretrained on out-of-task datasets (see the response above, `Fair Comparison with Vanilla Fine-Tuning of Backbone Model`).
> - Despite the simplicity of our method, our global reward inherently captures lookahead characteristics, which are crucial in offline RL (including MCTS) for avoiding local optima. This aligns with the main contribution of ILQL, which highlights that “our offline RL method can lead to significant improvements in final performance as compared to such ‘single step’ approaches, particularly when the training data is highly suboptimal for the desired task.” Additionally, we provide a comparison with the local-optimal (single-step) method, i.e., logitfusion [1], in Table 7.
>
> Reference:
>
> [1] Tuning language models by proxy
>
>
> We hope this pointwise comparison with ILQL addresses your concerns.
>
> Please let us know if you would like a more detailed comparison on other aspects.

---

> ### Author Response · Authors · 2024-12-02
>
> Dear Reviewer msmv,
>
> The discussion that we can participate in will end soon. Could you kindly confirm whether our responses have appropriately addressed your concerns? If you find that we have properly addressed your concerns, we kindly request that you consider adjusting your initial score accordingly. Please let us know if you have further comments.
>
> Thank you for your time and effort in reviewing our work.
>
> Best, Authors

---

### Official Review · Reviewer_TvLG · 2024-11-04

**Soundness:** 1
**Presentation:** 2
**Contribution:** 1
**Rating:** 5
**Confidence:** 4

**Summary:**

The paper proposes an approach to do faithful rationale generation in LLMs. It uses a steering-based approach to make the outputs more faithful to the reasoning of the llm in classification. The idea is to weight token logits using 2 kinds of reward models: A "local" one that tries to match tokens to those suggested by a domain-specific expert model and a "lookahead" one that does an MTCS type search and re-weights logits based on rewards from unrolled sequences.

Experiments are performed on a couple of QA type datasets, demonstrating that each method makes improvements in classification accuracy and faithfulness of rationales. Some qualitative analyses are also presented.

**Strengths:**

1. MTCS type inference is a hot topic right now, and it is indeed an important frontier for LLMs to improve on.
2. At a surface level, experimental results seem to show large gains.

**Weaknesses:**

Section 3 is pretty badly written, it is pretty hard to get the details of the approach. Instead of invoking irrelevant sophisticated-sounding terminology like "Feynman-Kac" formulas it would be better to describe the method in more detail. The math especially is confusing, see below.

The paper seems to show some positive experimental results, but I am concerned about whether we are looking at a meaningful comparison. The proposed methods rely on domain experts. Looking at table 8 in the appendix these are generally models that have been fine-tuned for the task in some way (and not just on the validation sets as the main section claims, some have access to external datasets). So it shouldn't be that surprising that a method that is given access to an expert which has more signal will do better than the backbone pre-trained model. A fair comparison would have to be with an approach that does vanilla fine-tuning of LLama or mixtral model.

In terms of novelty: The authors have not really cited relevant work in the controlled decoding space:

https://arxiv.org/abs/2310.17022

https://sea-snell.github.io/ILQL_site/

These works already do something more sophisticated than just token reweighting by a reward score. So what is the novel contribution here? 2 possibities:
1. Focusing on the faithfulness problem.
2. The "lookahead" idea of the reward model. I dont recall having seen this before, but it feels like a simplification of a full-blown MCTS. I would also call this a poor man's version of ILQL.

So we are just left with #1 then, unless I missed something. And this is something I consider of limited novelty (more like an application for a particular problem, though one with interesting implications from the steering perspective).

**Questions:**

1. what is "t \wedge T" ?
 2. sec 3.3, what is P(s_t) a posterior over?
 3. In what sense is \pi_t a "potential" function?
 4. I cannot make any sense of eq 2. Is w \in V the same as w_t?  why cant you simply remove the indicator function and write it as \sum_w \in C ? why is the indicator function in the deminator as well? is the intent to have a logit distribution that only puts mass on the tokens in C?
 5. are the rollouts done on the backbone model or the expert model? have we considered /measured the inference time cost? this is an important consideration in a paper about mtcs type methods.

6. Does q_\phi simply reward completions of the output that have tokens in C?

7. intro: "in contrast, an expert model....." : this is an interesting claim (does seem plausible). is there a citation for evidence?

8. line 140: tend to generate similar token....": what does this mean?

9. i am not up to date on the faithfulness literature, but the kind of interventations that the paper describe as standard ways of evaluation i.e. word inclusion and perturbation just seem to be likely to be noise-prone, leading to unreliable evals?

10. GenExpert =?  lookahead?

11. comment: the discussion between 329-342 helped understanding a bit and should be earlier in the paper.

12. sec 5.2.2: the NLI example is a bad one i think. Submergible only means it is something that can be submerged. which doesnt automatically mean it is submerged.

---

> ### Author Response · Authors · 2024-11-22
> **Response to reviewer TvLG (1)**
>
> Thank you for your valuable time and your thoughtful feedback! We address each point as follows.
>
> **Writing about method**
>
> Thanks for your thoughtful questions regarding the method details. We have updated the method section (Section 3.3 and Section 3.4) with a newly updated **Algorithm1** that elaborated on the pipeline, addressing your suggestion that *it would be better to describe the method in more detail*.
> - The **notation $t \wedge T$** represents the minimum of the two values $t$ and $T$. This mathematical convention is common in contexts like stochastic processes and optimisation. Here, it indicates that the product runs from $i = 1$ up to the lesser of $t$ and $T$. This effectively caps the sequence or product based on the minimum value.
> - **$\mathbb{P}_t​(s_t​)$** represents the probability of reaching $s_t$ from the $\mathbb{P}_t$. Specially, $[S_t=s_t]$ is an indicator function that is equal to 1 if the state at $t$ is $s_t$, and 0 otherwise. The numerator inside the expectation represents the product of rewards and the probability of reaching state $s_t$, ensuring that paths leading to high rewards over time are given more weight.
> - **Rollout in MCTS** is used to estimate the value of future actions, helping navigate and expand the search tree effectively. The Eq.(1) represents a probabilistic reward distribution of state $s_t$. $G(s_t,s_{t+1})$ is analogous to reward function in MCTS, with superiority in estimate the reward via lookahead. In our framework, the global expert model $U^g$ performs reward estimation to the generated policy from backbone model (see `GlobalReward` function).
> - **Inference efficiency**: we totally agree that inference-time cost is an important consideration in MCTS-like methods. Unlike existing *explicit* MCTS requiring expensive rollouts or simulations to evaluate potential actions, we compute expected rewards in a more integrated way, as shown in the numerator of Eq.(1), streamlining the decoding process. Our cost is similar to beam-search, while detailed comparison results is available in Table 14 (Table13 in original submission).
> -  **Equation about reward calculation**: to ease understand, we remove the Eq.2 and Eq.3, instead, we update function *LocalMask* and *GlobalReward* in **Algorithm 1** in Section 3.3.
>    - Local Reward: We introduce a set of classification label words and we remove the label words which are not included in the expert model’s prediction. We then renormalise the output probability based on the new vocabulary (See `LocalMask` function in Algorithm 1).
>    - Global Reward:  The expert model generate a lookahead reward by evaluating the plausibility of the tuple $(x_t, x_{t+1})$ generated from the backbone model (See `GlobalReward` function in Algorithm 1).

---

> > ### Author Response · Authors · 2024-11-22
> > **Response to Reviewer TvLG (2)**
> >
> > **Supported literature**
> >
> > **Q1**: Literatures about “expert model can generate domain-specific words”
> >
> > **R1**: We totally agree that evidence from existing literature about this claim will make our argument more supportive. In general, Domain-Specific Language Models are fine-tuned or trained from scratch on domain-specific data, enabling them to comprehend and generate language that reflects  the unique terminology, jargon, and linguistic patterns prevalent in that domain. These capabilities lead to better performances in domain-specific tasks, such as machine translation [4], Science[5] text understanding and question answering in biology [6]. More direct evidence in [7] shows that after training on the domain-specific corpus the masked language model loss decreases on 50K randomly sampled held-out documents from each domain, implying a better fit to the domain-specific text.
> >
> > Moreover, we calculated the percentage of generated domain-specific words ourselves and the results on ASAP dataset, i.e., student essay assessment. Sepfically, we use TF-IDF to select the top 200 words from the prompt including question, key elements and rubric, as the domain-specific words. Then, we calculate the percentage of these domain-specific words in the responses from backbone model, expert model and our model (all of them are llama3-8b).
> > The results over science(Q1, Q2) and biology(Q3, Q4) subjects shown below clearly verify that the expert model responds to the question with more domain-specific words.
> >
> > ||Q1|Q2|Q3|Q4|
> > |---|---|---|---|---|
> > |Backbone|14.36%|6.85%|0.07%|1.04%|
> > |Expert|16.88%|19.26%|0.26%|1.12%|
> >
> > **Q2**: line140, the instruction-tuned model generate similar token distribution in [1]
> >
> > **R2**: The original sentence in [1] is “Surprisingly, we find that base and aligned LLMs (e.g., Llama-2 and Llama-2-chat) typically perform almost identically in most positions in terms of ranking tokens during decoding”. That is, given the same input context, the token distributions produced by these models are similar at any position in the generated sequence. In their evaluation across 1,000 examples, 92.2% of the tokens overlapped between the base and aligned LLMs. We reference this observation to emphasise that even advanced (instruction-tuned) models face limitations in generating knowledge-intensive words.
> >
> > **Q3**: Faithfulness evaluation based on perturbation
> >
> > **R3**: Yes, the state-of-the-art faithfulness evaluation methods are based on perturbation-based methods. For example,[2] introduces mistakes or biases into the context, while [3] evaluates faithfulness by removing subsets of input tokens.
> >
> > **Novelty**
> >
> > Thanks for highlighting the two related works. Both primarily focus on training strategies,  whereas our method is implemented during inference. Your insights into the two possibilities are absolutely correct. Our constrained generation framework is specially designed for faithful rational generation, employed a simplified yet more efficient version of MCTS. Despite its simplicity, our contributions are not trivial:
> > - As demonstrated in Table 1, we first identified the importance of domain-specific words in enhancing context adherence. This insight motivated us to increase the generation probability of domain-specific words, leading to the use of a global reward (from the expert model) to improve faithfulness. To the best of our knowledge, this is the first study to improve faithfulness by explicitly encouraging the generation of domain-specific tokens.
> > - Since there is a trade-off between faithfulness and accuracy [2], we incorporate a local reward derived from an expert classification model to enhance accuracy before generating rationales.
> > - From technical perspectives, most MCTS-based methods typically contributes on designing task-specific rewards and efficiently incorporating them into the simulation process. Our method is different from your ILQL in at least two key aspects: (i) we proposed two novel reward mechanisms tailored for the faithfulness problem (ii) our simulation process integrates both the expert model and the backbone model. Our implicit MCTS-framework avoids the explicit rollout process while preserving the core principles of MCTS that (a) estimating node rewards based on lookahead $G_t$. (ii) facilitating a more principled framework to balance exploration and exploitation through the probabilistic expectation, rather than rely on a balance coefficient.
> >
> > Above all, our contribution extends far beyond simply adapting a weaker version of an existing method to a particular task. Our framework can be applied to other constrained generation tasks, such as personalised generation and diversity-oriented generation, by defining different rewards, while keeping low computation costs.
> >
> > We have updated the **Contribution in Section 1** and the **Comparison with existing MCTS-like decoding methods** in Section 3.3, line 167-172, to highlight the novelty of our framework.

---

> ### Author Response · Authors · 2024-11-22
> **Response to Reviewer TvLG (3)**
>
> **Fair Comparison with Vanilla Fine-Tuning of Backbone Model**
>
> Thank you for pointing out this critical point. The comparison results are updated in Table 5 and Table 5, where **CLS** is the expert model used to predict the answer and the **expert model** refers to the global expert model is used to generate rationale. As CLS is a classifier that can't generate rationale, we only apply it in acc evaluation, while expert model is applied in both metrics calculation. Below is our analysis:
>
> **(a) comparing with fine-tuned backbone model**:
>    We have updated the ablation results in Table 5 and Table 6. For the ASAP dataset, the expert model uses the same backbone as the primary model (i.e., Llama3-8B) and has been fine-tuned on the ASAP training set. This aligns well with your intended comparison with *Vanilla Fine-Tuning of Backbone Model*
>    - **Accuracy Results**: In Table 5, by comparing the expert model with our full framework, we observe overwhelming advantages of our full model across **all four subsets**, especially on Q2, with results showing 68% vs. 48% (ours vs. expert). And the average results are 74\% and 69\% for our (full) and expert model.
>    - **Faithfulness Results**: The expert-only method achieves better faithfulness on three subsets (except for Q4). Interestingly, our full framework (including CLS & expert) behaves better than our+expert model, showing that the synergised effects from CLS and expert.
> - **Overall Results**: Although our method fails to exhibits better faithfulness compared with the fine-tuned model expert model, it instead strikes a challenging trade-off between accuracy (CLS) and faithfulness (expert model), which has been discussed in [2]. This is also one of the core motivation: combining the strengths of classification-specialised and rationale-specialised models.
>
> **(b) compare with smaller fine-tuned models**:
> For other datasets where the expert models are Llama2-7B (smaller than our backbone): (i) The comparison between our full (i.e., *our+cls+expert*) and *our+expert* also show that the synergistic effects of combining the two experts. (ii) the faithfulness for our (full) is better than expert model model on SNLI and MNLI, with 15\% vs 13\% and 19\% vs 9\%, respectively. This suggests that our approach not only achieves faithfulness improvements over smaller expert models but also has the potential to leverage weak supervision to unlock the capabilities of larger backbone models.
>
> **(c) Generalised to Out-of-Task Expert Model**:
> Noted that it is not strictly necessary for the expert model to be trained on the exact task dataset. For instance, we experimented with Expert Model 2 (/Weyaxi/Einstein-v2-7B in huggingface), which is trained on general science question-answering instead of ASAP-specific data. Results in Table 12 show that incorporating this out-of-task expert model leads to improved faithfulness on 11 out of 12 metrics. These results validate the generalisability of our method.
>
> Overall, our framework demonstrates: (i) superior performance in balancing accuracy and faithfulness when the expert model is of the same size, (ii) improved faithfulness compared to smaller, in-task trained expert models, and (iii) robust generalizability, effectively adapting to scenarios where the expert model has not been specifically trained for the inference task.
>
> **Case study**
>
> We appreciate your observation that “submergible” does not necessarily mean the individual is submerged. Our key point here is to emphasise that our method can faithfully respond to meaningful changes in the input (even if the response is imperfect), whereas the backbone model completely ignores the perturbation.
>
> For better clarity, we replace an example shown below (also updated in the revised pdf):
> ``` Perturbed Premise: [frugally Requires free registration.
> Hypothesis: Does not require free registration.
> Backbone: entailment; Requires free registration is a necessary condition for only if Requires free registration.
> Ours: Contradiction; The premise states that the website [frugally] requires free registration, which implies that a user must provide some information or sign.
> ```
>
>
> **References**:
>
> [1] The unlocking spell on base LLMs: Rethinking alignment via in-context learning. ICLR2024
>
> [2] Question Decomposition Improves the Faithfulness of Model-Generated Reasoning. ICML23
>
> [3] On Measuring Faithfulness or Self-consistency of Natural Language Explanations. ACL24
>
> [4] Fine-tuning Large Language Models for Domain-specific Machine Translation
>
> [5] SciBERT: A Pretrained Language Model for Scientific Text. ACL2019
>
> [6] BioBERT: A Pretrained Biomedical Language Representation Model for Biomedical Text Mining. Bioinformatics 2020.
>
> [7] Don’t Stop Pretraining: Adapt Language Models to Domains and Tasks. ACL 2020
>
>
> Please let us know if you have further concrete questions or concerns that we can address. Thank you for your engagement with our work.

---

> ### Author Response · Authors · 2024-11-25
>
> We've taken your initial feedback into careful consideration and incorporated them into our manuscript as indicated in our response. Could you kindly confirm whether our responses have appropriately addressed your concerns? If you find that we have properly addressed your concerns, we kindly request that you consider adjusting your initial score accordingly. Please let us know if you have further comments.
>
> Thank you for your time and effort in reviewing our work.

---

> ### Author Response · Authors · 2024-12-02
> **Response to Reviewer TvLG: Follow-up Questions**
>
> As the discussion period is nearing its end, we kindly request your feedback on the comparison with ILQL provided below. We hope it addresses your concerns adequately.
>
> *We followed the thread of your feedback and noticed it was mistakenly placed in the "Reviewer msmv" block*. To ensure clarity, we have moved our response here. We apologise for any inconvenience this may have caused and sincerely appreciate your attention to this matter.
>
> Firstly, at a high level, both methods share the similarity of leveraging logit perturbation to enhance the output of the backbone model. For clarity, we denote the backbone model as $M_b$ and the expert model as $M_p$.
>
> Then, we provide a detailed comparison in the following aspects:
>
> **Model Types**
> - ILQL: Both the backbone model (referred to as the standard language model) and the perturbed model are GPT-2 small.
> - Ours: Our backbone models are LLaMA3-8B and Mistral-7B, while perturbed models include LLaMA3-8B, LLaMA2-7B, and Mixtral-7B. Importantly, our method supports backbones and perturbed models of differing sizes and tokenisations.
>
> **Does it require extra training before deployment?**
> - ILQL: Yes, there are three types of $M_{p}$: policy generation $\pi_B$, value function model for $Q$ and $V$ generation, and a target value network. They are trained via implicit Q-learning objectives on the exact dataset same as inference tasks.
> - Ours:  No, we can use any $M_{p}$ pretrained on in-task datasets, or even out-of-task datasets (see response above `Fair Comparison with Vanilla Fine-Tuning of Backbone Model`.)
>
> **Perturbation strategy**:
> - ILQL:  calibrate the original policy using trained $Q$ and $V$, i.e., $\pi(a \mid h) \propto \pi_\beta(a \mid h) e^{\beta(Q(h, a) - V(h))} = \exp(\log(\pi_\beta(a \mid h)) + \beta(Q(h, a) - V(h))).$
> - Ours: the step-wise reward is given by $M_p$, and then select the sequence with highest sentence-level reward from generated $K$ sequences (see in Algorithm 1). This reward strategy is inspired by our preliminary study in Figure 1 that $M_p$ prefers in-domain text and could contribute to faithfulness.
>
> **Overall analysis**
> - Our method can fit the situation where $M_b$ and $M_p$ have different but larger backbones. In contrast, ILQL's reliance on training three models for $M_p$ can limit its scalability to larger backbone models, such as fine-tuning a LLaMA3-8B. Our approach, however, can leverage any expert model, even if pretrained on out-of-task datasets (see the response above, `Fair Comparison with Vanilla Fine-Tuning of Backbone Model`).
> - Despite the simplicity of our method, our global reward inherently captures lookahead characteristics, which are crucial in offline RL (including MCTS) for avoiding local optima. This aligns with the main contribution of ILQL, which highlights that “our offline RL method can lead to significant improvements in final performance as compared to such ‘single step’ approaches, particularly when the training data is highly suboptimal for the desired task.” Additionally, we provide a comparison with the local-optimal (single-step) method, i.e., logitfusion [1], in Table 7.
>
> Reference:
>
> [1] Tuning language models by proxy
>
>
> We hope this pointwise comparison with ILQL addresses your concerns.
>
> Please let us know if you would like a more detailed comparison on other aspects.

---

### Author Response · Authors · 2024-11-22
**Summary of revision in pdf**

Thank you very much for all the reviewers' feedback. We have explicitly incorporated your initial feedback into our revised PDF (in blue):

- Summarised our three-fold **contributions in the Introduction.**
- Revision on **method introduction:**
  - **Algorithm Description In Section 3.3**: we introduced a step-by-step description of our method in Algorithm 1.
  - **Feynman-Kac Model Explanation in Section 3.3**: we reorganised the introduction of the model and added a more thorough explanation of Eq. (1).
  - **Local and Global Reward Formulations**: we elaborate how the rewards are calculate by referring to Algorithm 1.
- Update the **ablation study results** We updated the performances of vanilla expert models in **Table 6**.
- Replace the **case for NLI in Section 5.3 Case Studies.**
- Add detailed **evaluation** and **hyper-parameter** in **Appendix A.**
- Add **faithfulness evaluation results** on three tasks based on the **Mistral-7B backbone in Appendix B.**
- In **Appendix C.1**, we elaborated on how our framework can be generalized to tasks where the **answer space is unconstrained.**

---

### Author Response · Authors · 2024-11-22
**General response to all reviewers**

We thank all reviewers for their valuable feedback and dedicated time! As some of the reviewers are concerned about the method details, here we give a brief explanations about our motivation and method:

**Motivation of leveraging the expert model for faithful rationale generation**

As demonstrated in Table 1, we first identified the importance of domain-specific words in enhancing context adherence. This insight motivated us to increase the generation probability of domain-specific words, leading to incorporation of expert model trained on specific domains for faithfulness enhancement. To the best of our knowledge, this is the first study to improve faithfulness by explicitly encouraging the generation of domain-specific tokens.

**The advantages of our overall probabilistic inference framework by comparing with other MCTS-like methods**

Our faithfulness-seeking model is techinically distinguished in *computation efficiency* and *lookahead rewards*.
- Unlike existing explicit MCTS [1,2] requiring expensive rollouts or simulations to evaluate potential actions, it computes expected rewards in a more integrated and efficient way, streamlining the inference process.
- The incorporated lookahead is achieved on cumulative rewards across multiple steps, rather than overly prioritising short-term gains or rely on heuristics, e.g., the normalised average that do not model future states effectively.

**Reward**

**Local reward**: Inspired by the literature that domain-specific experts tend to demonstrate better accuracy in knowledge-rich tasks. Specifically, we introduce a set of classification label words and penalise label words that are not included in the expert model’s predictions. The output probabilities are then renormalised to ensure validity.

**Global reward**:  As a faithful rationale requires coherence with the surrounding context. We use a expert model that is trained on domain-specific corpus to provide **lookahead** reward when generating rationales. Specifically, the expert model scores the current state $x_t$ by evaluating the generated $(x_t, x_{t+1})$ from backbone LLMs. It is expected that text spans faithful and coherent with the domain-sensitive context are preferred, as they better align with the expert's fine-tuned distribution.

**Contribution and novelty**

- We investigate the challenge of faithful rationale generation by highlighting the limitations of general LLMs in producing domain-specific responses. To the best of our knowledge, this is the first study to enhance faithfulness by explicitly encouraging the generation of domain-specific tokens.
- We propose two novel reward mechanisms, namely local and lookahead rewards, tailored for the rationale generation problem. These are integrated into a efficient probabilistic inference framework to achieve a trade-off between task accuracy and rationale faithfulness.
- Empirical results show both enhancement in accuracy and faithfulness over seven tasks, with an absolute accuracy improvement of 33\% over the seven datasets, along with a 10\% improvements in faithfulness evaluation, while maintaining a computation cost similar to beam search (1.3$\times$).

References:

[1] Don’t throw away your value model! generating more preferable text with value-guided monte-carlo tree search decoding.

[2] Pairwise optimization for o1-like olympiad-level mathematical reasoning.

---

### Note · Authors · 2024-12-15

I have read and agree with the venue's withdrawal policy on behalf of myself and my co-authors.